# Stereospecific lasofoxifene derivatives reveal the interplay between estrogen receptor alpha stability and antagonistic activity in *ESR1* mutant breast cancer cells

David J Hosfield[1], Sandra Weber[2], Nan-Sheng Li[1], Madline Sauvage[2], Carstyn F Joiner[3], Govinda R Hancock[3], Emily A Sullivan[3], Estelle Ndukwe[1], Ross Han[1], Sydney Cush[1], Muriel Lainé[1], Sylvie C Mader[2], Geoffrey L Greene[1], Sean W Fanning[3]*

[1]Ben May Department for Cancer Research, University of Chicago, Chicago, United States; [2]Institute for Research in Immunology and Cancer, Université de Montréal, Montréal, Canada; [3]Department of Cancer Biology, Loyola University Chicago, Maywood, United States

*For correspondence:
sfanning@luc.edu

**ABSTRACT** Chemical manipulation of estrogen receptor alpha ligand binding domain structural mobility tunes receptor lifetime and influences breast cancer therapeutic activities. Selective estrogen receptor modulators (SERMs) extend estrogen receptor alpha (ERα) cellular lifetime/accumulation. They are antagonists in the breast but agonists in the uterine epithelium and/or in bone. Selective estrogen receptor degraders/downregulators (SERDs) reduce ERα cellular lifetime/accumulation and are pure antagonists. Activating somatic *ESR1* mutations Y537S and D538G enable resistance to first-line endocrine therapies. SERDs have shown significant activities in *ESR1* mutant setting while few SERMs have been studied. To understand whether chemical manipulation of ERα cellular lifetime and accumulation influences antagonistic activity, we studied a series of methylpyrollidine lasofoxifene (Laso) derivatives that maintained the drug's antagonistic activities while uniquely tuning ERα cellular accumulation. These molecules were examined alongside a panel of antiestrogens in live cell assays of ERα cellular accumulation, lifetime, SUMOylation, and transcriptional antagonism. High-resolution x-ray crystal structures of WT and Y537S ERα ligand binding domain in complex with the methylated Laso derivatives or representative SERMs and SERDs show that molecules that favor a highly buried helix 12 antagonist conformation achieve the greatest transcriptional suppression activities in breast cancer cells harboring WT/Y537S *ESR1*. Together these results show that chemical reduction of ERα cellular lifetime is not necessarily the most crucial parameter for transcriptional antagonism in *ESR1* mutated breast cancer cells. Importantly, our studies show how small chemical differences within a scaffold series can provide compounds with similar antagonistic activities, but with greatly different effects of the cellular lifetime of the ERα, which is crucial for achieving desired SERM or SERD profiles.

## Editor's evaluation

This work elegantly provides novel insights into the molecular mechanisms of selective Estrogen Receptor modulator and downregulator activities on Estrogen Receptor function.

## Introduction

The pro-oncogenic cellular activities of estrogen receptor alpha (ERα) drive breast cancer pathogenesis. ERα is overexpressed in approximately 70% of breast cancers and targeted endocrine therapies are given to prevent primary disease metastasis. Post-menopausal patients primarily receive aromatase inhibitors, which indirectly inhibits ERα by ablating endogenous estrogens, and may be given in combination with a CDK 4/6 inhibitor (*Burstein et al., 2019*). Pre-menopausal patients receive tamoxifen, a selective estrogen receptor modulator (SERM), which directly binds to the receptor and reprograms transcription to induce cellular quiescence (*Maximov et al., 2018*). Together, these therapeutic paradigms significantly reduce the 5 years risk of recurrence and continually show improved disease outcomes (*Early Breast Cancer Trialists Collaborative Group, 2005*; *Mouridsen et al., 2003*).

Acquired resistance to endocrine therapies remains a major source of breast cancer mortality. Approximately 50% of patients present acquired or de novo resistance to endocrine therapies after an average of 5-years (*Anurag et al., 2018*). Deep genomic sequencing of endocrine-resistant, ER+, and metastatic breast cancers revealed the presence of *ESR1* ligand binding domain mutations (*ESR1*$_{muts}$) at a rate of approximately 25% (*Toy et al., 2013*; *Jeselsohn et al., 2014*; *Robinson et al., 2013*). Y537S (14%) and D538G (36%) are the two most prevalent mutations and account for nearly 50% of those identified. Both mutations, especially Y537S, enable hormone-free transcriptional activity, resistance to inhibition by 4-hydroxytamoxifen (4OHT, the active metabolite of tamoxifen), and are associated with a more aggressive disease phenotype (*Chandarlapaty et al., 2016*). *ESR1*$_{muts}$ have also been identified in cultured breast cancer cells and become enriched as they become resistant to endocrine therapies, suggesting a latent and selectable mechanism of acquired drug resistance (*Martin et al., 2017*). In addition to mimicking the genomic actions of E2 stimulation, *ESR1*$_{muts}$ bind to unique cistromes and promote allele-specific transcriptional programs compared to E2-stimulated WT receptor (*Jeselsohn et al., 2018*). Mutation-induced molecular alterations enable resistance to clinically approved antiestrogens and a more aggressive metastatic progressive disease.

Biochemical studies and x-ray crystal structures have shown that both mutations favor an E2-like agonist conformation in the absence of hormone (*Fanning et al., 2016*; *Nettles et al., 2008*). This favored agonist conformation reduces the binding affinity of ERα ligands and alters the therapeutically important antagonist receptor conformation of the SERM 4OHT (*Fanning et al., 2016*). Competitive antiestrogens with selective estrogen receptor degrader/downregulator (SERD) activities with improved potencies show superior therapeutic antagonistic activities compared to 4OHT in breast cancer cells harboring Y537S or D538G *ESR1* (*Fanning et al., 2018b*). The SERD fulvestrant, contrary to 4OHT, completely ablated Y537S ERα transcriptional activity in breast cancer cells (*Toy et al., 2013*). Fulvestrant's antiestrogenicity stems from its ability to antagonize ERα transcription and reduce its cellular lifetime (degrade/downregulate receptor) by inducing post-translational modifications including SUMOylation and ubiquitination by increasing proteasomal degradation (*Wijayaratne and McDonnell, 2001*; *Traboulsi et al., 2019*; *Hilmi et al., 2012*). Tamoxifen and other SERMs also antagonize transcription but induce a stable antagonist conformation that enhances ERα nuclear lifetime (*Fanning et al., 2018b*). SERMs show partial agonist activities in the uterine endometrium and/or in bone while SERDs are pure antagonists (*Fanning and Greene, 2019*). Due to unfavorable pharmacological properties of fulvestrant, large-scale efforts brought about the development and clinical evaluation of improved SERDs (*Guan et al., 2019*; *Wardell et al., 2015b*; *Hamilton, 2018*; *Tria et al., 2018*; *Dickler, 2018*; *Bardia et al., 2019*; *Wardell et al., 2017*; *Metcalfe et al., 2019*; *Fanning et al., 2018a*). These molecules possess functional groups including carboxylic acids, azeditines, and pyrrolidines that favor the H12 antagonist conformation and induce proteasomal degradation by increasing its conformational mobility, although their impact on receptor post-translational modifications remains uncharacterized (*Fanning and Greene, 2019*; *Guan et al., 2019*; *Fanning et al., 2018a*; *De Savi et al., 2015*). Interestingly, recent studies suggest that ERα degrading activities may not be required for therapeutic efficacy in breast cancer cells with recurring hotspot *ESR1* mutations (*Fanning et al., 2018b*; *Lainé et al., 2021*; *Wardell et al., 2015a*; *Andreano et al., 2020*). In light of this recent work, the role of ERα modification by ubiquitin and SUMO, cellular lifetime, and accumulation in anti-cancer therapeutic activities, especially with the *ESR1*$_{muts}$, remains unclear.

In this study, we developed a series of novel lasofoxifene (Laso) derivatives that affect ERα cellular accumulation similar to either SERMs or SERDs, providing a comparable chemical scaffold to study the

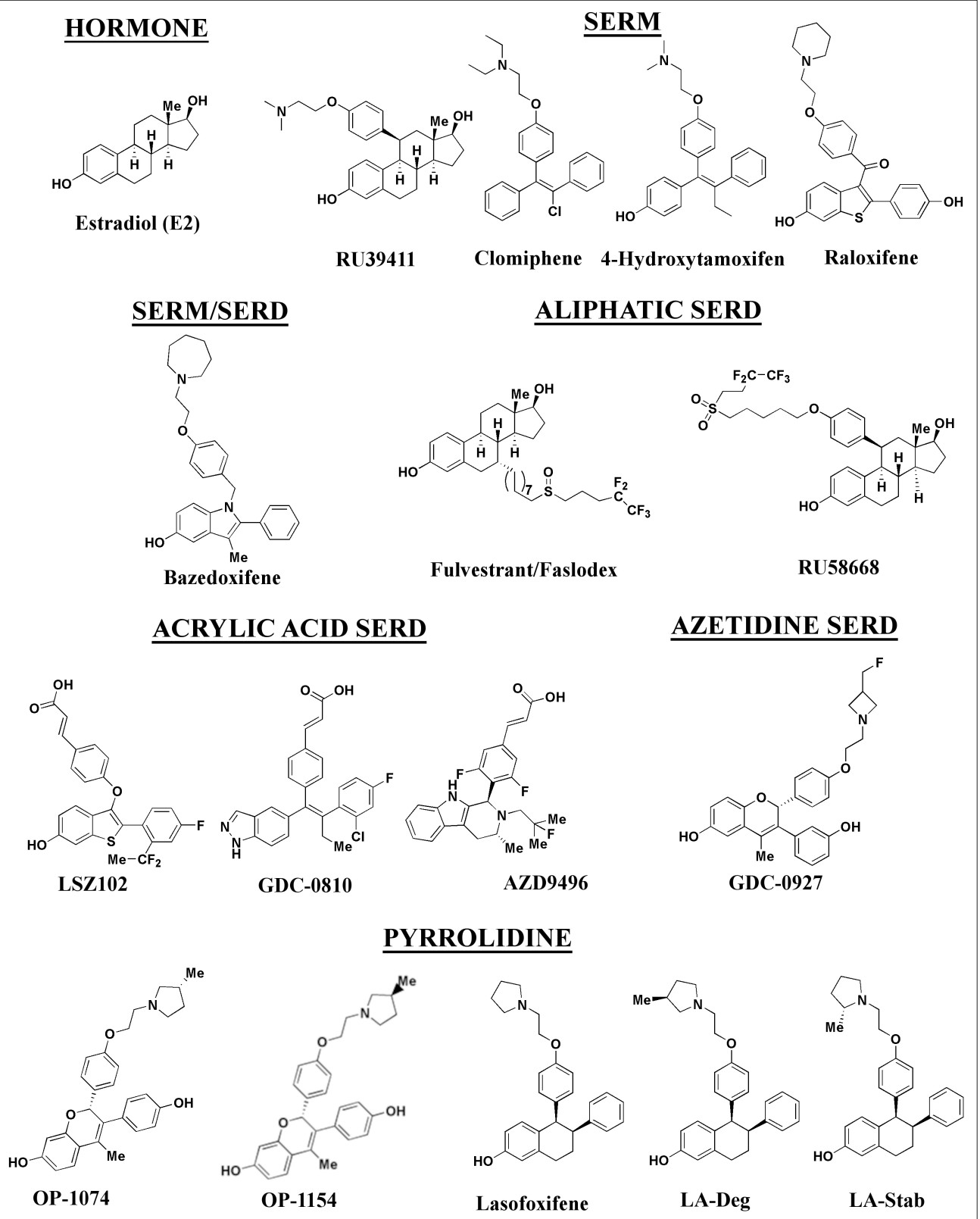

**Figure 1.** Estrogen receptor alpha ligands evaluated in this study including estradiol, selective estrogen receptor modulators (SERMs), and selective estrogen receptor degraders/downregulators (SERDs).

role of receptor stability on transcriptional antagonistic and antiproliferative activities. We evaluated these molecules against a panel of SERMs and SERDs (*Figure 1*) representing a comprehensive spectrum of clinical and preclinical antiestrogenic small molecules. As part of these studies, we developed a novel high-throughput live-cell approach to quantify and observe the kinetics of ERα cellular accumulation within living breast cancer cells upon treatment with SERMs or SERDs. Because fulvestrant-induced SUMOylation of WT ERα correlates with shortened kinetics of interaction with DNA, lowered accessibility of estrogen response elements, and more efficient repression of ERα target genes in the presence of fulvestrant (*Traboulsi et al., 2019*; *Hilmi et al., 2012*), bioluminescence resonance energy transfer (BRET) was used to examine differences between SERM and SERD-induced SUMOylation for WT, Y537S, and D538G ERα. Reporter gene assays quantifying the impact of SERM or SERD treatment on WT transcriptional antagonism in WT or heterozygous Y537S/WT and D538G/WT MCF7 breast cancer cells revealed that antiestrogen efficacy at suppressing transcription did not necessarily correlate with reduced receptor accumulation. Further, cellular proliferation studies in MCF7 with heterozygous WT/Y537S *ESR1*, which possesses the greatest degree of antiestrogen-resistance, show that manipulating ERα stability did not enhance anti-proliferative activities in cultured cells. Comparison of Y537S ERα LBD x-ray crystal structures in complex with SERMs and SERDs reveal a ligand-dependent H12 conformational pose that corresponds to increased transcriptional antagonistic efficacies.

## Results

### Ligand influence on ERα expression in live breast cancer cells

We developed a live cell-based assay to quantify differences in ERα expression and cellular lifetime based on ligand or mutation. T47D breast cancer cells were engineered to express a doxycycline (dox) inducible halo-tagged (HT) ERα, allowing the cells to be treated with a permeable HT-specific fluorophore (halo-618) and levels of ERα expression quantified following the addition of dox. To determine ERα accumulation, cells were distributed into 96-well plates in estrogen depleted medium. Using a range of dox concentrations, we found that 1 μg/mL was the lowest dose needed to achieve maximal signal-to-noise ratio (*Figure 2—figure supplement 1*). To uncover whether ligands and activating somatic mutations influenced the cellular lifetime of ERα, we simultaneously induced halo-ERα expression, treated with 1 μM ligand, and measured fluorescence normalized to cell count every 4 hr for 100 hr. *Figure 2* shows representative curves plotting the mean fluorescence counts of two biological replicates (± SD) after normalization to cell count by phase contrast. For WT cells, ERα reached a maximum at approximately 24 hr then fell to baseline by 80 hr (*Figure 2A*). Treatment with E2 lead to a reduced expression but the signal converged with vehicle by around 30 hr and had fully returned to baseline around 80 hr. Treatment with SERDs further reduced the signal and led to a rapid return to baseline, at around 45 hr for fulvestrant (*Figure 2A*). Treatment with SERMs greatly enhanced ERα levels over vehicle and extended the time to baseline to past 80 hr (*Figure 2D*). Interestingly, veh-treated WT and Y537S cells showed similar profiles, but the Y537S mutant took slightly longer to reach baseline (*Figure 2B*). Unlike the E2-treated WT cells, E2-treated Y537S cells showed an identical pattern to veh-treated Y537S cells (*Figure 2B*). Surprisingly, the Y537S mutant extended the ERα lifetime in the presence of fulvestrant, past 60 hr compared to 45 hr for WT (*Figure 2B*). Unexpectedly, the D538G mutant not only reduced the magnitude of ERα expression, but also the kinetics of its cellular turnover with a greater time to baseline for each SERD (*Figure 2C*). For SERMs, each molecule showed similar profiles of enhancing ERα lifetime in the WT T47D cells (*Figure 2D*). In the 537 S cells, they further enhanced stability compared to WT (*Figure 2E*). Surprisingly, RU39411 was the only SERM to elicit any kind of difference on D538G by showing a slight increase in stability; however, it was still approximately 2.5-fold less than in the Y537S (*Figure 2F*). These data suggest that, in addition to hormone independent transcriptional activities, both Y537S and D538G mutations uniquely affect ERα cellular lifetime within T47D cells.

### Y537S and D538G mutations uniquely affect the potency and efficacy of ERα degradation

To quantitate ERα stabilization or degradation/downregulation by different ligands, cells were simultaneously treated by dox, halo-618 and hormone (E2), SERMs, SERM/SERDs, or SERDs at concentrations

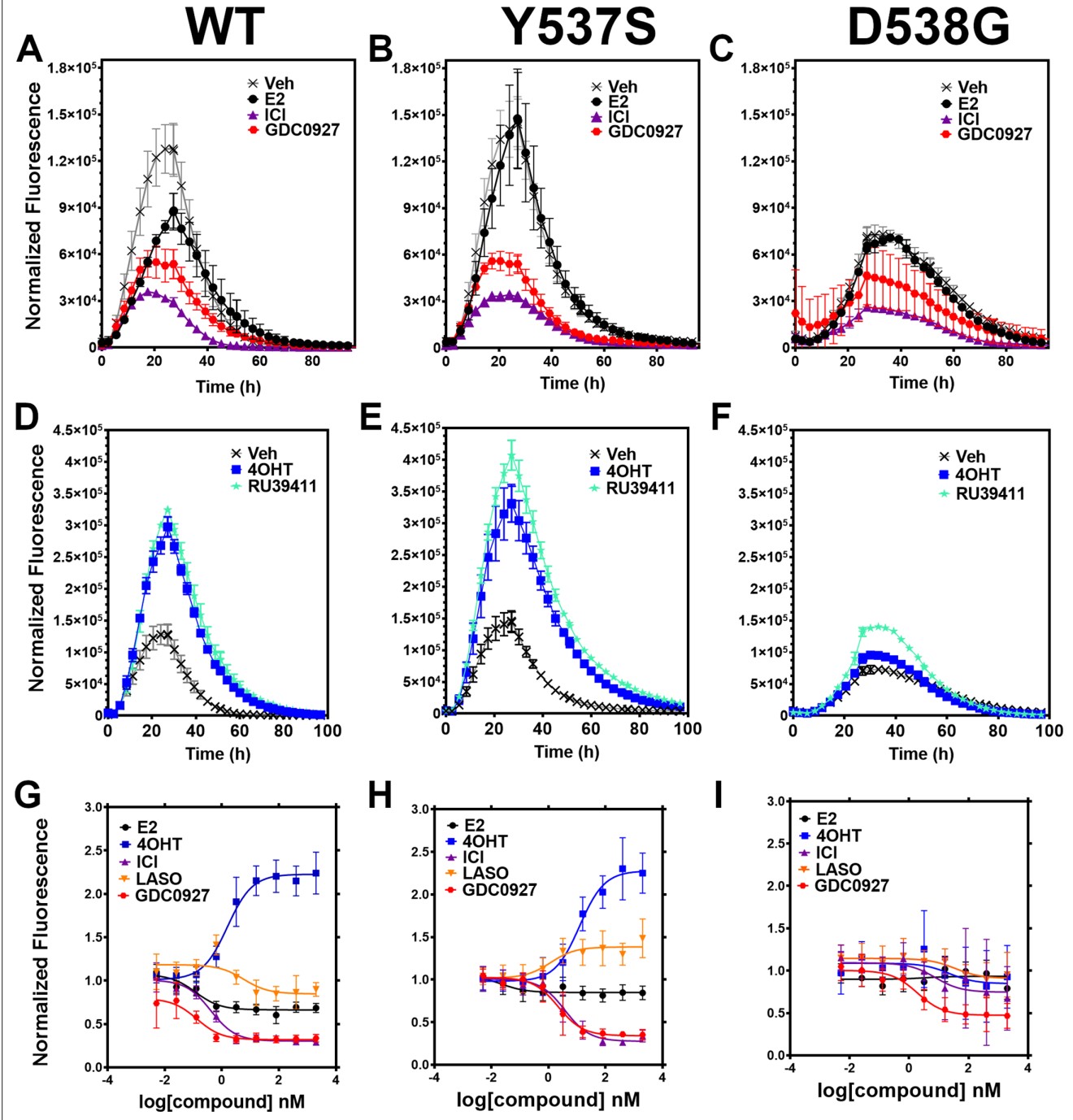

**Figure 2.** Impact of ligand and mutation on Halo-estrogen receptor alpha (ERα) lifetime in T47D breast cancer cells. (**A–C**) Halo-618 fluorescence measured every 4 hr in T47D cells expressing WT halo-ERα (**A**), Y537S (**B**), and D538G (**C**) treated over 100 hr with vehicle (Veh), 1 μM estradiol (E2), fulvestrant (ICI), or GDC0927 following induction of expression. (**D–F**) Same conditions as in (**A–C**), except that cells were treated with Veh, 4-hydroxytamoxifen (4OHT) or RU39411. Data are normalized to cell count in each well and are shown as the mean of two biological replicates ± SD (**G–I**) TMR signal in T47D breast cancer WT (**G**), Y537S (**H**), or D538G (**I**) ERα treated for 24 hr with between 2.5 pm and 1 μM E2, 4OHT, ICI, lasofoxifene (Laso), or GDC0927. All data are normalized to vehicle and are shown as the mean of two biological replicates ± SD.

The online version of this article includes the following source data and figure supplement(s) for figure 2:

**Source data 1.** Ligand and mutational influences on estrogen receptor alpha (ERα) cellular turnover after 24 hr.

**Figure supplement 1.** Doxycycline induction of Halo-ERα after 24 hr.

**Figure supplement 2.** IC$_{50}$ of fulvestrant (ICI) in normal T47D cells.

**Figure supplement 3.** In-cell western of T47D breast cancer cells treated for 24 hr with 4-hydroxytamoxifen (4OHT) or fulvestrant (ICI).

varying between 0.1 nM and 5 µM for 24 hr. *Figure 2—source data 1* shows $IC_{50}$, R (*Maximov et al., 2018*) of fit, and fluorescence levels normalized to cell count at maximum dose (5 µM) for each ligand and ERα mutant. As expected, treatment with SERMs resulted in increased maximal signals, while SERDs induced a significant decrease (*Fanning et al., 2018b*). SERMs, such as 4OHT, show a normalized fluorescence greater than 1 at 5 µM, whereas Laso was neutral (between 0.8 and 1), and SERDs reduced fluorescence levels to less than 0.8. As expected, the hormone E2 induced ERα degradation, which at 0.68 ± 0.05 was approximately 2–3-fold less than for the SERDs with the greatest reduction, at 0.2 ± 0.03, 0.29 ± 0.04, and 0.33 ± 0.04 for RU58668, fulvestrant (ICI182,780), and GDC-0927 respectively. The SERMs 4OHT and RU39411 significantly increased ERα signal with a normalized fluorescence of 2.23 ± 0.24 and 2.11 ± 0.36 at 5 µM treatment respectively. Overall, GDC-0927, fulvestrant, and pipendoxifene (PIP) showed the most potent $IC_{50}$ values at 0.13 ± 0.03, 0.4 ± 0.02, and 0.77 ± 0.03 nM. Notably, many oral SERDs were not significantly different from PIP. To validate our data, we treated normal T47D cells with increasing doses of fulvestrant then used a capillary-based western blotting technique to observe differences in ERα levels after 24 hr normalized to β-actin. We derived an $IC_{50}$ value of 0.34 ± 0.10 with an R (*Maximov et al., 2018*) of 0.94 using this method, which is in close agreement with the results from the engineered HT-ERα expressing cells (*Figure 2—figure supplement 2*). We also used an in-cell western approach to validate these data for endogenous ERα in T47D cells and observed similar trends in receptor levels after 24 hr (*Zhao et al., 2017*; *Figure 2— figure supplement 3*).

To understand how Y537S and D538G ERα mutations impact the stabilizing or degrading activities of antiestrogens, we generated stable T47D breast cancer cells that possess dox-inducible HT Y537S and D538G ERα. SERM-like molecules appeared to retain their abilities to increase receptor nuclear lifetime in the Y537S, but did so with approximately 10-fold reduced $IC_{50}$ values (*Figure 2H* and *Figure 2—source data 1*). Laso, which was essentially neutral to slightly SERD-like in the WT cells, showed a SERM-like profile with Y537S (1.48 ± 0.22 at 5 µM). Interestingly, SERD molecules that possessed carboxylic acids (AZD9496, LSZ102, and GDC0801) all showed more SERM-like profiles with their normalized fluorescence above 1 at 5 µM. Fulvestrant, RU58668, GDC-0927, and PIP were the only molecules that did not show statistically significant differences to their maximal normalized fluorescence with Y537S. However, they did show about a 10-fold decreased $IC_{50}$ compared to WT. Surprisingly, introduction of the D538G mutation mostly abolished ligand-dependent control on receptor levels (*Figure 2I*). Nearly every tested SERM and SERD had decreased $IC_{50}$ values and the greatly increased maximum dosages to achieve an effect on ERα levels (*Figure 2—source data 1*). GDC-0927 was the only SERD that attained maximal effect on normalized fluorescence with an $IC_{50}$ <10 nM at 1.88 ±0.04 nM.

## Stereo-specific addition of methyl groups to Laso's pyrrolidine tunes ERα cellular lifetime

A major goal of this study was to understand whether chemical manipulation of WT and mutant ERα cellular lifetime enhanced transcriptional antagonistic activities. Laso did not significantly affect ERα signal compared to vehicle across all concentrations. Notably, Laso possesses a pyrrolidine moiety, similar to OP-1074, which induces ERα degradation through a 3R-methylpyrrolidine (*Fanning et al., 2018a*). Hence, we synthesized a series of Laso derivatives with stereospecific methyl groups at the 2 and 3 positions on the pyrrolidine (*Figure 3A*). We first examined the abilities of the molecules to influence WT halo-ERα lifetime in our engineered T47D cells (*Figure 3B*). LA-3 possesses a 3 R methylpyrrolidine (similar to OP1074) and showed a SERD-like profile with an $IC_{50}$ of 26.94 ± 0.4 nM and maximum normalized fluorescence of 0.48 ± 0.07 at 5 µM. LA-5 possesses a 2 S methylpyrrolidine and showed a SERM-like profile with an $IC_{50}$ of 15.68+0.27 nM and a normalized fluorescence of 1.706 ± 0.09 at 5 µM. LA2 and LA4 appeared similar to Laso in that they did not affect receptor lifetime.

LA3 and LA5 were synthesized as racemic mixtures. To understand whether one chiral species accounted for the activity, we performed chiral separation and were able to resolve two major peaks (*Figure 3—figure supplement 1*). Both LA3 and LA5 showed significantly improved $IC_{50}$ values for the second peak versus the first peak (*Figure 3C*). For LA3 peak 1 (LA3-1) and LA3-2 the $IC_{50}$ was 112.9 ± 0.23 nM and 2.58 ± 0.13 nM respectively. The normalized fluorescent signals at 5 µM were somewhat decreased for LA3-1 versus LA3-2 at 0.57 ± 0.02 and 0.49 ± 0.02 respectively. They were also slightly decreased at 1.75 ± 0.02 and 1.66 ± 0.04 for LA5-1 and LA5-2 respectively. As such, chiral separation

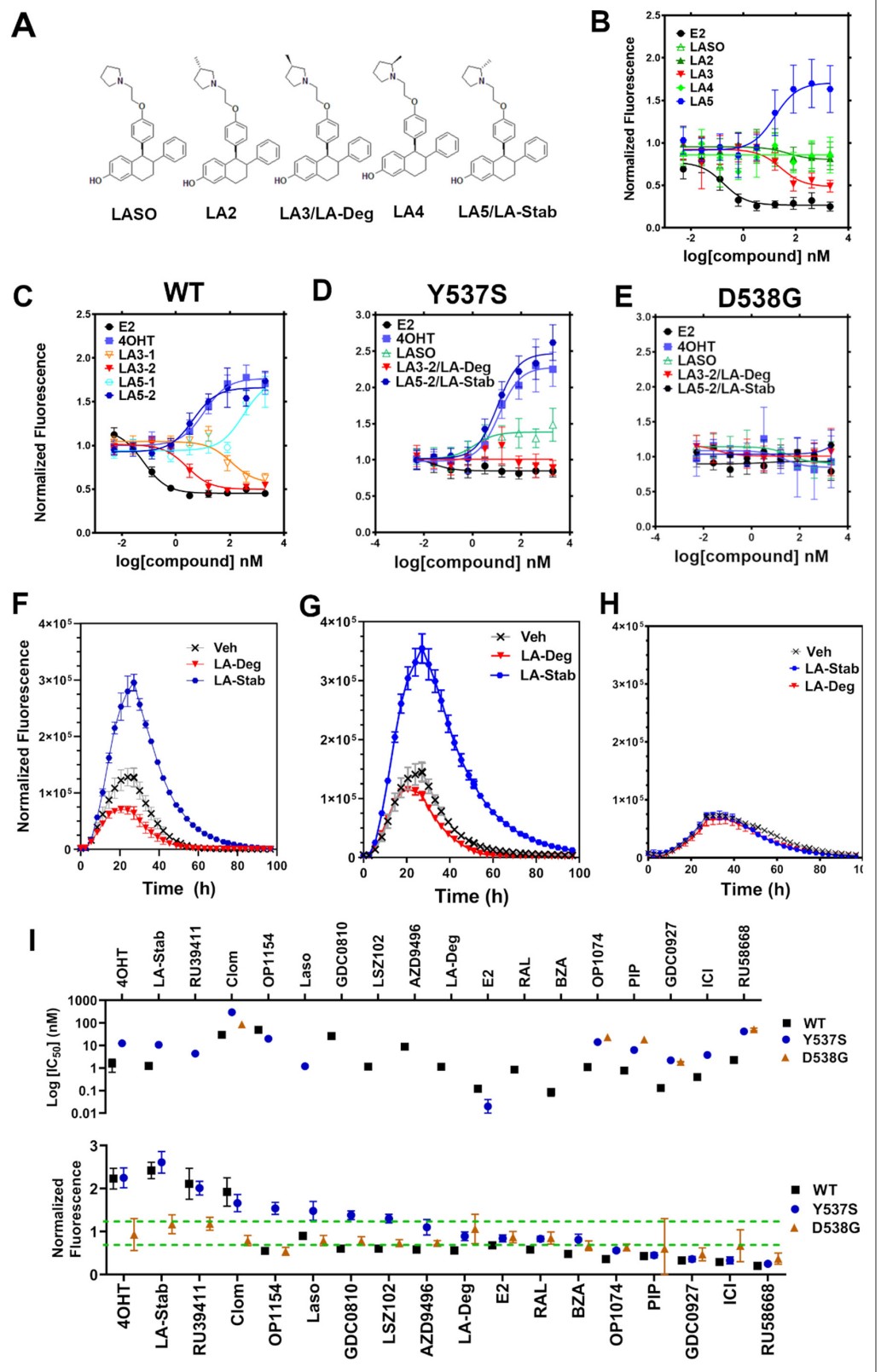

**Figure 3.** Stereospecific methyl additions onto the pyrrolidine of lasofoxifene (Laso) impact estrogen receptor alpha (ERα) levels in T47D breast cancer cells. (**A**) Chemical structures of Laso and the synthesized stereospecific methyl derivatives. (**B**) Dose-response curves of hormone (**E2**) alongside LASOLaso and derivatives after 24 hr treatment for WT halo-ERα. (**C**) Dose-response curves of chirally purified LA3 and LA5 alongside E2 and

*Figure 3 continued on next page*

*Figure 3 continued*

4-hydroxytamoxifen (4OHT) for WT halo-ERα. LA3/5-1 and LA3/5-2 represent the first and second major peaks separated by chiral affinity chromatography. (**D/E**) Dose-response curves of LA-Deg and LA-Stab for Y537S and D537G halo-ERα after 24 hr compared to E2, Laso, and 4OHT. (**F–H**) TMR fluorescence measured every 4 hr in T47D breast cancer cells with WT halo-ERα (**F**), Y537S (**G**), and D538G (**H**) treated over 100 hr at 1 μM LA-Stab or LA-Deg following induction of expression. Data are shown as the mean of two biological replicates ± SD.

The online version of this article includes the following figure supplement(s) for figure 3:

**Figure supplement 1.** Chiral separation of 3 R (LA-Deg) and 2S-methylpyrrolidine (LA-Stab) lasofoxifene derivatives.

**Figure supplement 2.** In-cell western of T47D breast cancer cells treated with lasofoxifene (Laso).

substantially improved the $IC_{50}$ but not the fluorescence at maximum dose. LA3-2 and LA5-2 are subsequently referred to as LA-Deg and LA-Stab for Laso degrader and stabilizer respectively. We used an in-cell western approach to validate these data for endogenous ERα and observed similar trends in receptor levels after 24 hr (*Figure 3—figure supplement 2*).

We next examined the abilities of LA-Deg and LA-Stab to affect the cellular lifetime of Y537S and D538G halo-ERα in T47D breast cancer cells. In Y537S cells, LA-Stab demonstrated a slightly improved $IC_{50}$ over 4OHT at 10.62 ± 0.12 and 12.42 ± 0.15 nM respectively. LA-Stab also increased Y537S levels to a slightly greater extent than 4OHT at maximum dose at a signal of 2.47 ± 0.06 and 2.27 ± 0.07 respectively (*Figure 3D*). Interestingly, the Y537S essentially neutralized the SERD activities of LA-Deg with a normalized fluorescence of 1.01 ± 0.04 at 5 μM. Laso showed a more SERM-like profile with a normalized fluorescence of 1.38 ± 0.03 at 5 μM. As with other molecules, the D538G mutant completely neutralized the LA-Deg and LA-Stab with normalized fluorescent signals around 1 at 5 μM treatment. In the kinetics/lifetime assays, the methylated Laso derivatives showed similar profiles compared to other antagonists (*Figure 3F,H*). Summary data for $IC_{50}$ and normalized fluorescence at maximum dose are shown for all antagonists in *Figure 3I*. Overall, these data show that

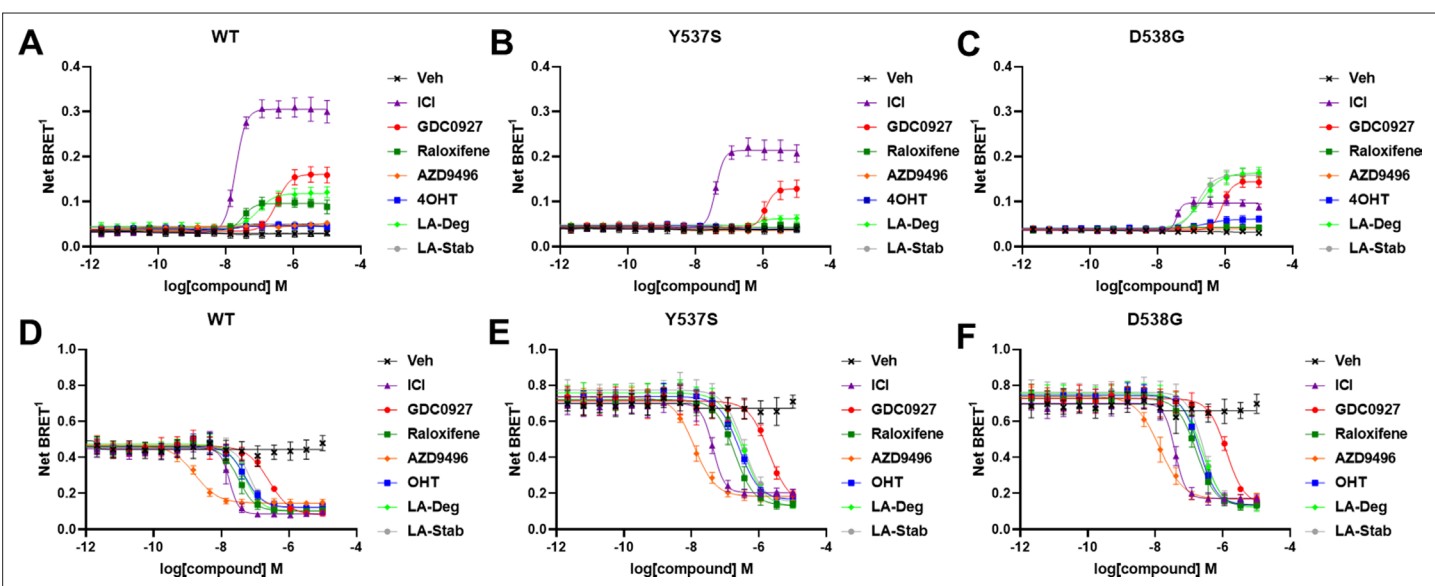

**Figure 4.** Impact of Y537S and D538G mutation on ligand-induced estrogen receptor alpha (ERα) SUMOylation and SRC1 coactivator binding. SUMOylation of WT (**A**), Y537S (**B**), or D538G (**C**) ERα in the presence of vehicle, fulvestrant, GDC0927, raloxifene, AZD9496, 4-hydroxytamoxifen (4OHT), lasofoxifene-degrader (LA-Deg), or lasofoxifene-stabilizer (LA-Stab). Data are shown as the mean ± SEM n=3–5 biological replicates. Association of WT (**D**), Y537S (**E**), and D538G (**F**) ERα and the receptor-interacting domain of SRC1. Data are shown as the mean ± SEM, n=3 biologic replicates.

The online version of this article includes the following source data and figure supplement(s) for figure 4:

**Source data 1.** Ligand and Mutational Influences on Estrogen Receptor SUMOylation.

**Source data 2.** $IC_{50}$ of SRC1 receptor interacting domain binding to WT and mutant estrogen receptor alpha.

**Figure supplement 1.** Fulvestrant-induced SUMOylation WT.

the methylated Laso derivatives have different impacts on WT ERα accumulation and lifetime but are affected by Y537S and D538G mutations similar to other antiestrogens with comparable properties.

## Impact of ligands and mutations on SUMOylation

The activating somatic mutations Y537S and D538G differentially influenced how ligands affect ERα cellular lifetime. Fulvestrant induces both ubiquitination and SUMOylation of ERα in breast cancer cells, impacting its stability and activity (*Hilmi et al., 2012*). SUMOylation of ERα can be measured in HEK293T cells using BRET between Renilla Luciferase (RLucII) tagged ERα and YFP tagged SUMO3 (*Hilmi et al., 2012*; *Cotnoir-White et al., 2018*). *Figure 4* shows representative SUMO BRET response curves for WT, Y537S, and D538G ERα with a panel of antiestrogens. *Figure 4—source data 1* shows $IC_{50}$ values and maximum BRET ratios for the SUMOylation experiments. In both WT and Y537S cells, fulvestrant induces the greatest degree of SUMO3 recruitment, but the $IC_{50}$ values and maximum BRET ratios are decreased with the Y537S mutant. GDC0927 is the next most efficacious antiestrogen and its potency trends similarly to fulvestrant with the Y537S mutant. LA-Deg induces SUMOylation but to a lesser maximum than fulvestrant and GDC0927 in WT and is further reduced in Y537S, while LA-Stab did not induce SUMOylation with either receptor. Interestingly, RAL, which showed weak SERD-like activity in WT but not mutant ERα in our cellular lifetime and accumulation assays, also induced low levels of SUMOylation in the WT but not the Y537S cells. Overall, these assays show a correlation between a molecule's ERα-degrading activities and induction of SUMOylation. However, the acrylic acid SERD AZD9496 did not induce SUMOylation with any ERα construct.

As in our cellular lifetime studies, the D538G mutant showed the greatest alterations in induced SUMOylation compared to WT and Y537S ERα in the BRET assay. Fulvestrant induced about 25% of the SUMOylation for D538G compared to WT ERα. GDC0927 was less affected in its capacity to induce SUMOylation by D538G, albeit its potency was reduced, consistent with our lifetime/accumulation experiments. Surprisingly, LA-Stab and LA-Deg displayed similar activity, contrary to observations with WT and Y537S ERα. Interestingly, 4OHT also induced low levels of SUMOylation in the D538G mutant that were not observed for WT and Y537S. These gains in SUMOylation for LA-Stab and 4OHT correlate with their loss of stabilization activity observed in T47D cells. Because fulvestrant showed significant differences in SUMOylation for the D538G mutant, we examined whether addition of E2 further affected its efficacy for induction of this post-translational modification (*Figure 4—figure supplement 1*). Both the efficacy and potency of fulvestrant for SUMOylation were markedly reduced, similar to our previous studies in which they were significantly reduced for the D538G mutant. Together, these studies show that the D538G activating somatic *ESR1* mutation uniquely affects receptor SUMOylation in response to SERDs, in agreement with our measured differences in cellular lifetime.

## Hotspot mutations enhance coactivator recruitment and reduce inhibitory potencies in cells

Ligand-dependent transcriptional activities depend on the association of coactivator proteins with ERα to form functional complexes (*Liao et al., 2002*). SERMs and SERDs elicit a structural rearrangement of the ERα LBD to sterically preclude binding of coactivators in the activating function-2 cleft (*Fanning and Greene, 2019*). Mutations affect the ligand-dependence, antagonistic efficacies and potencies of SERMs and SERDs, and specific coactivator proteins that associate with ERα (*Jeselsohn et al., 2018*; *Fanning et al., 2016*; *Fanning et al., 2018b*). As such, we used BRET to understand whether SERD molecules showed improved abilities to inhibit the binding of the receptor-interacting domain (RID) of the coactivator SRC1 with WT and mutant ERα in cells. RLucII tagged ERα and YFP tagged SRC1 RID were used to measure differences in association in HEK293T cells. *Figure 4D* shows representative curves for WT and mutant ERα-SRC1 RID binding. *Figure 4—source data 1* shows $IC_{50}$ values and maximum BRET ratios for the coactivator binding experiments. Overall, enhanced basal BRET ratios are observed for the mutants compared to WT ERα. The mutations also decreased the potencies of all SERMs and SERDs to inhibit coactivator binding, consistent with earlier findings for 4OHT and bazedoxifene (BZA) using purified recombinant proteins in vitro (*Fanning et al., 2018b*). Interestingly, all molecules showed similar efficacies in inhibiting this interaction at the greatest concentrations. AZD9496 showed the greatest potency for all ERα species, but is approximately 10-fold reduced in the mutants. Fulvestrant was the next most potent. Raloxifene, 4OHT, LA-Deg, and LA-Stab showed

similar potencies in cells with WT and mutant ERα. GDC0927 showed the least potent inhibition of coactivator association in the WT and mutant ERα assays. Because SERDs showed varied potencies in prohibiting coactivator association with the ERα mutants and did not display greater efficacies, we conclude that induction of ERα SUMOylation does not correlate with repression of SRC1 RID recruitment in cells. Importantly, the mutants uniquely affect which coactivator proteins associate with ERα (*Jeselsohn et al., 2018*). Therefore, additional studies will be needed to identify preferred antiestrogens to target mutant-specific cofactor interactions.

## SERMs and SERDs antagonize *ESR1* mutant transcriptional activities

We used reporter gene assays to measure transcriptional antagonistic potencies of antiestrogens in breast cancer cells harboring WT, Y537S, or D538G *ESR1*. A 3x-estrogen response element DNA sequence upstream of a GFP reporter was stably incorporated into MCF7 cells harboring WT, heterozygous WT/Y537S, or heterozygous WT/D538G *ESR1* (donated by Dr. Ben Ho Park). Cells with stable incorporation of the gene cassette were selected for and enriched using flow sorting. Cells were placed in serum-starved media for 48 hr then treated with increasing concentrations of ligand. ERα transcription was determined by quantitating the total integrated green fluorescence of these cells 24 hr later (*Figure 5*, *Figure 5—source data 1*). For WT cells: GDC0927, RAL, and OP1074 showed the most potent transcriptional inhibition at $IC_{50}$ = 0.03 ± 0.08, 0.11 ± 0.08, and 0.14 ± 0.05 nM respectively, compared to 0.35 ± 0.06 nM for fulvestrant (*Figure 5D and G*). At the 5 µM maximum dose, RU39411, RAL, and BZA showed the greatest reduction in transcription at 0.14 ± 0.02, 0.16 ± 0.02, and 0.17 ± 0.01. In the WT/Y537S heterozygous cells GDC0927, RAL, and Laso showed the most potent inhibition at $IC_{50}$ = 0.95 ± 0.51, 2.16 ± 0.67, and 2.88 ± 0.34 nM respectively. At the maximum dose: RU39411, BZA, and PIP showed the greatest reduction in transcription at 0.33 ± 0.02, 0.32 ± 0.05, and 0.32 ± 0.04 respectively. As previously reported, no antiestrogen completely reduced transcriptional activity in cells that possessed Y537S ERα (*Toy et al., 2013*). In the WT/D538G heterozygous cells RU39411, fulvestrant, and OP-1074 showed the best $IC_{50}$ values at 0.26 ± 0.53, 0.57 ± 0.69, and 0.52 ± 0.63 nM respectively. At the maximum dose: OP1154, PIP, fulvestrant, and OP1074 showed the greatest transcriptional inhibition at normalized fluorescence of 0.25 ± 0.03, 0.29 ± 0.07, 0.32 ± 0.03, and 0.37±0.05 respectively. Only these four molecules returned the D538G transcriptional activity back to their respective WT values. Together, these data suggest that neither the potency nor the efficacy of induced ERα degradation correlate with improved transcriptional inhibition for the Y537S and D538G mutant receptors in this assay.

## LA-Stab induces alkaline phosphatase activity in uterine cells

Induction of alkaline phosphatase is an in vitro assay that correlates with ERα-mediated uterine stimulation, (*Holinka et al., 1986*) a hallmark of tissue-specific SERM partial agonist activities. We measured the abilities of E2, 4OHT, Z-endoxifen, ICI, Laso, LA-Stab, and LA-Deg to induce alkaline phosphatase in Ishikawa uterine epithelial cells using previously published methods (*Fanning et al., 2018a*). *Figure 6* shows the measured AP assay dose-response curves. 1 nM E2 induced the greatest AP response followed by LA-Stab, 4OHT, Z-endoxifen, and Laso. LA-Deg showed a slightly increase induction compared to ICI and Veh, but it was significantly lower than Laso. *Figure 6—source data 1* shows a table of the $EC_{50}$ and maximum absorbance for these assays. Together, these data suggest that the SERM-like effects on ERα accumulation observed for LA-Stab correlate to an increased stimulation of ERα-activities in uterine cells, while, modest ERα-degradation by LA-Deg was sufficient to prevent uterine stimulation.

## LA-Deg and LA-Stab show anti-proliferative activities in MCF7 cells with WT/Y537S *ESR1*

We used label-free cell counting to determine whether induction of ERα degradation improved anti-proliferative outcomes in MCF7 cells with heterozygous WT/Y537S *ESR1*. This model was chosen because it shows the greatest resistance to inhibition by antiestrogens (*Toy et al., 2013*). Cells were treated with 4OHT, fulvestrant, Laso, LA-Stab, or LA-Deg at 1 nM, 50 nM, and 1 µM in the presence of 1 nM E2 alongside vehicle (DMSO) and 1 nM E2-only controls. Subsequently, cells were counted every 6 hr for 84 hr. The experiment was halted when cells in the E2-treated well reached 100% confluence. All treatments were performed with three biological replicates and a total of 9 technical replicates

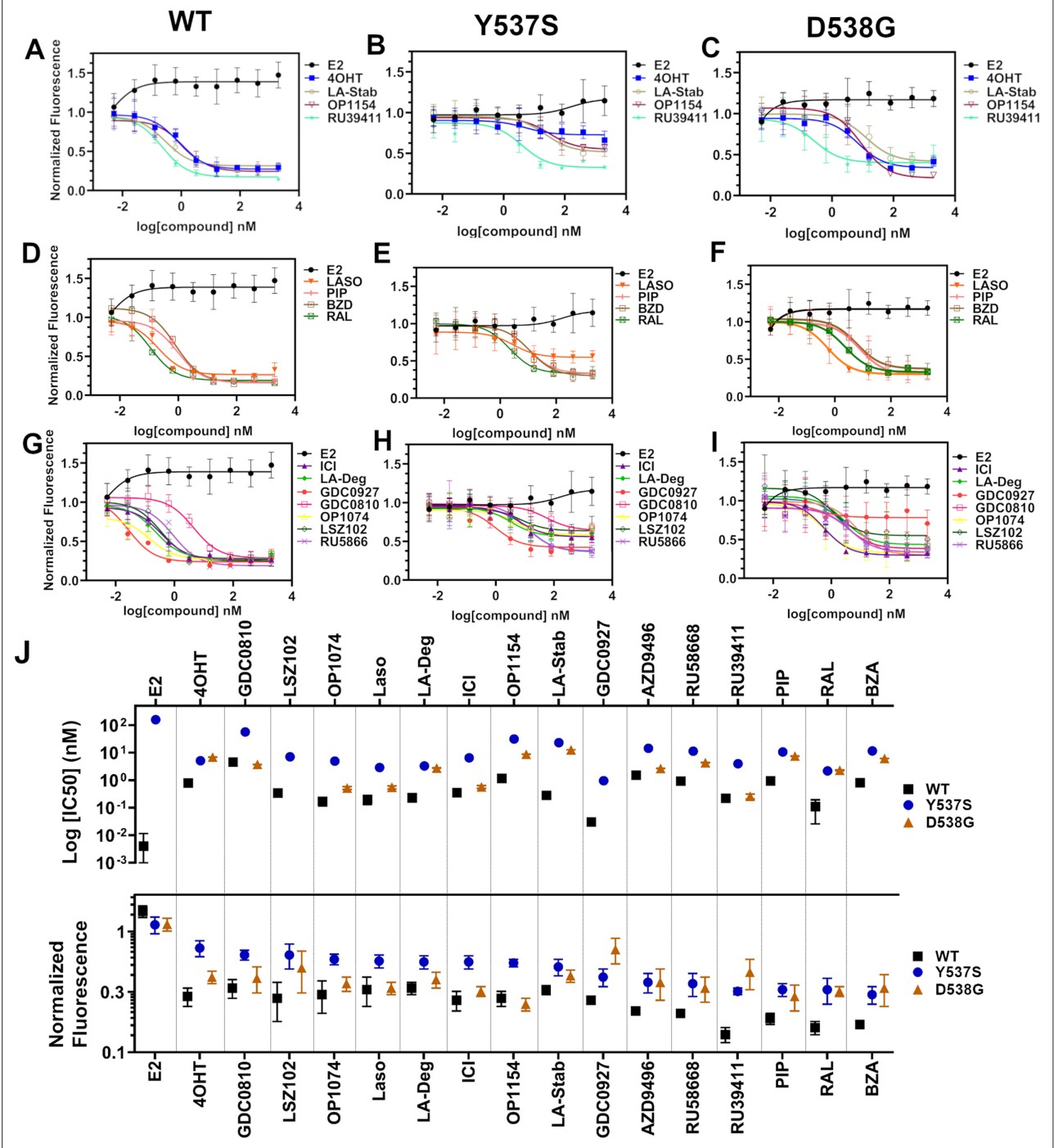

**Figure 5.** Transcriptional reporter gene assays in MCF7 cells with WT, WT/Y537S, and WT/D538G *ESR1*. Selective estrogen receptor modulators (SERMs) in (**A**) WT, (**B**) WT/Y537S, and (**C**) WT/D538G MCF7 cells. SERM/selective estrogen receptor degraders/downregulators (SERDs) in (**D**) WT, (**E**) WT/Y537S, (**F**) WT/D538G MCF7 cells. SERDs in (**G**) WT, (**H**) WT/Y537S, and (**I**) WT/D538G MCF7 cells. (**J**) Summary of $IC_{50}$ and normalized fluorescence at 5 µM compound after 24 hr for each compound. Data are ordered from left to right based on normalized fluorescence at highest dose in Y573S MCF7. Poor $IC_{50}$ fits were omitted. Data are shown as the mean of two biologic replicates ± SD. All data are normalized to cell count in their respective wells.

The online version of this article includes the following source data for figure 5:

**Source data 1.** Ligand and mutational influences on estrogen receptor alpha reporter gene transcription after 24 hr.

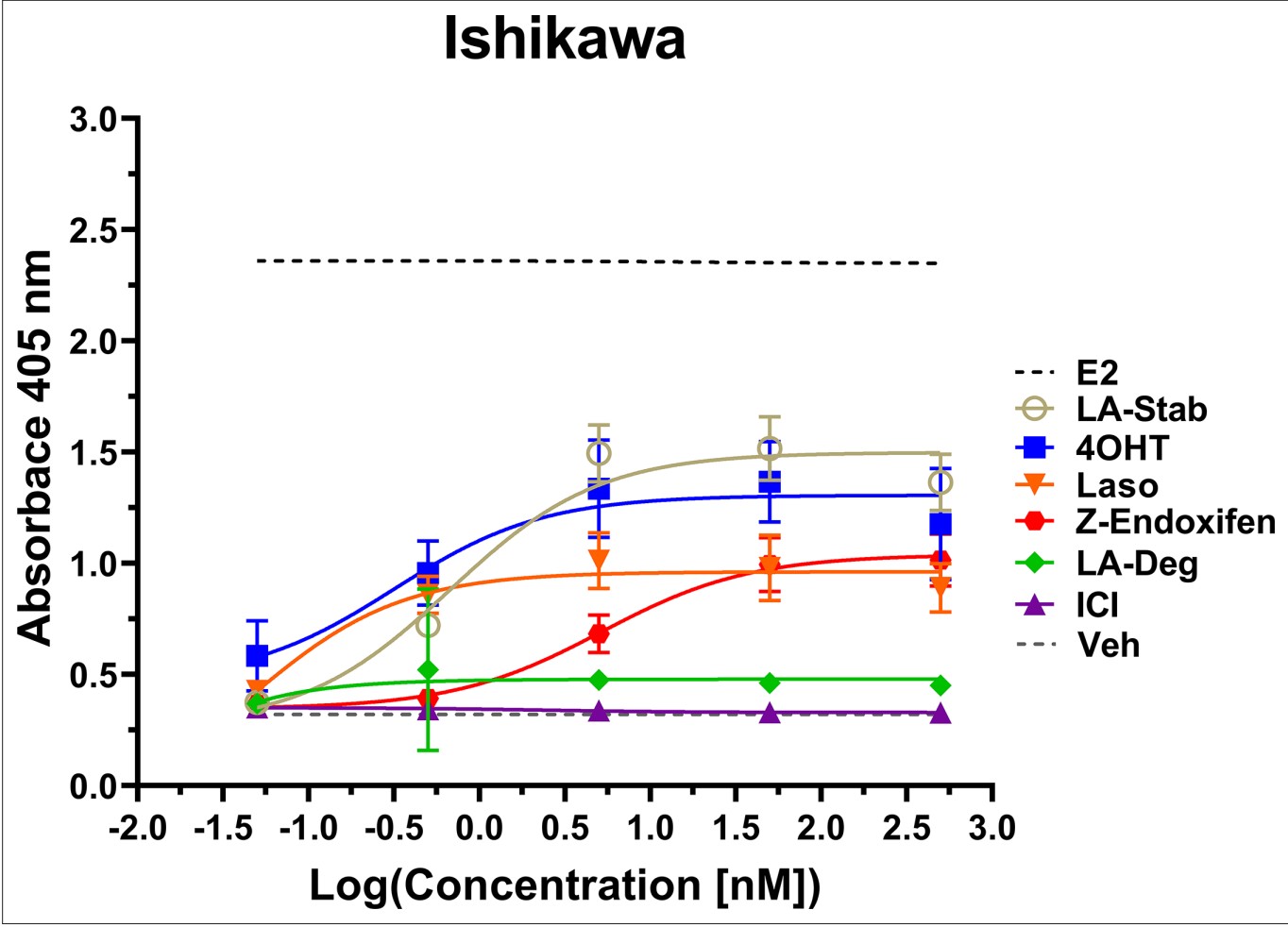

**Figure 6.** LA-Stab induces uterotrophic activity. Ishikawa cells treated with representative selective estrogen receptor modulators (SERM) or selective estrogen receptor degraders/downregulators (SERD) for 3 days in the absence of E2 and assayed for AP activity. Data shown are the mean ± SD. (n=3 independent replicates, 9 technical replicates total).

The online version of this article includes the following source data for figure 6:

**Source data 1.** Selective estrogen receptor modulator-agonist activities in uterine epithelial cells measured by induction of alkaline phosphatase.

per ligand per concentration. *Figure 7* shows the proliferation data and a summary of results. As expected, addition of 1 nM E2 enhanced cell proliferation. No compound completely abrogated proliferation, even at the highest dose. Every compound shows similar efficacies at the maximum 1 μM dose, although LA-Deg and LA-Stab appear slightly improved and closest to veh levels. However, differences were not statistically significant compared to the other molecules at the maximal dose. Fulvestrant and Laso on the other hand show improved efficacies at the 1 nM and 50 nM doses compared to the other molecules. Together, these data show that induction of ERα degradation does not by itself predict the anti-proliferative activities of an antiestrogen in this cell model.

### Structural basis of improved Y537S ERα antagonism

We solved x-ray crystal structures of antiestrogens in complex with Y537S and WT ERα ligand binding domain (LBD) to understand the structural basis of differential transcriptional antagonistic efficacies. We solved x-ray crystal structures for BZA, RAL, 4OHT, LA-Deg, LA-Stab, RU39411 and LSZ102 in complex with the Y537S mutant. We also solved structures with the WT LBD for those molecules with no existing structures (LA-Deg, LA-Stab, RU39411) or were able to solve to improved resolution (RAL). We were unable to solve x-ray crystal structures for antagonists in complex with the D538G mutant, despite earlier success with 4OHT (*Jeselsohn et al., 2018*). Overall, these structures present canonical

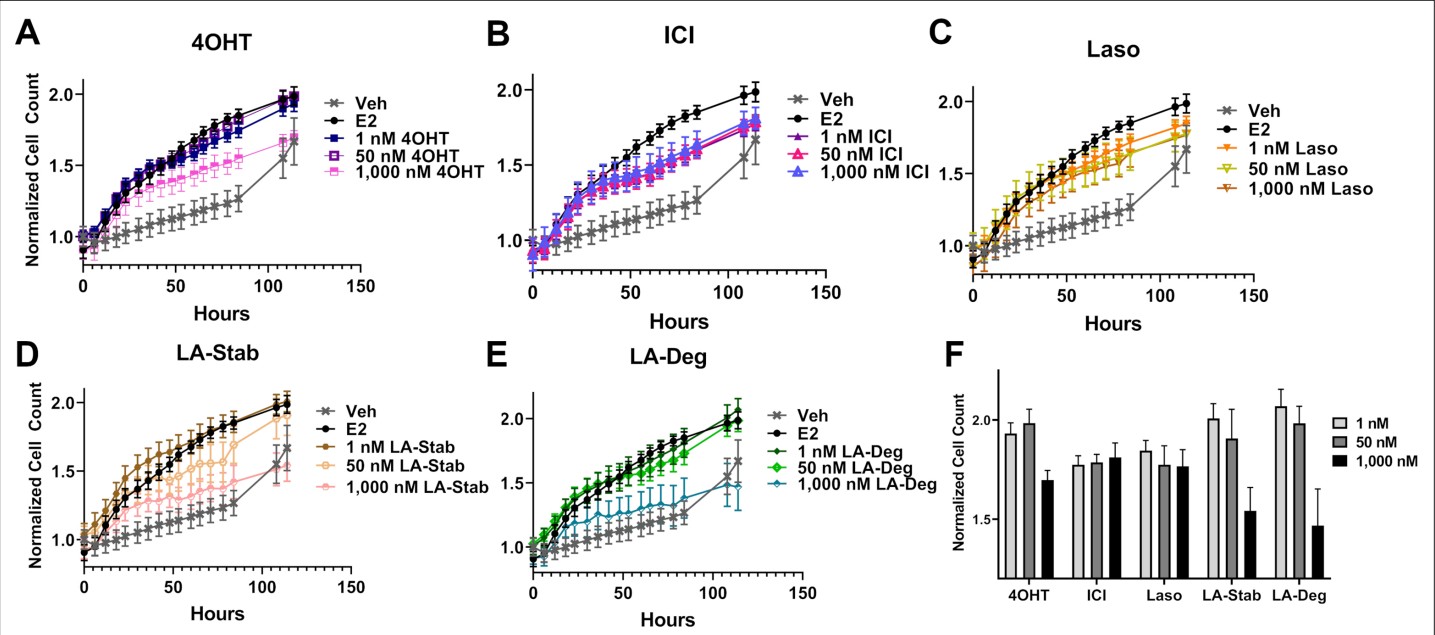

**Figure 7.** Anti-proliferative activities of selective estrogen receptor modulators (SERMs) and selective estrogen receptor degraders/downregulators (SERDs) in MCF7 cells with heterozygous WT/Y537S *ESR1*. (**A**) 4-hydroxytamoxifen (4OHT). (**B**) Fulvestrant (ICI). (**C**) Lasofoxifene (Laso). (**D**) Lasofoxifene-Stabilizer (LA-Stab). (**E**) Lasofoxifene-Degrader (LA-Deg). (**F**) Normalized cell count after 84 hr. All antagonist treatments are in the presence of 1 nM estradiol (E2). All data are normalized to initial cell count of the vehicle wells in their respective plates. All data are mean ± SEM for three biological replicates and a total of nine technical replicates.

ERα LBD homodimers with 2–4 monomers in the asymmetric unit. As with other ERα LBD structures, these show significant crystal contact influences at H12 of one monomer but not the other, namely a contact at or near 538 that affects the positioning of 537 (*Figure 8—figure supplement 1*). For analysis purposes, we have named the monomer free of H12 free of crystal contacts 'Chain A'.

Enforcing WT Antagonist Conformation Enables Efficacious Transcriptional Inhibition of Y537S ERα. RAL, BZA, and RU39411 showed the greatest efficacy of transcriptional inhibition in MCF7 cells with WT/Y537S *ESR1*. Laso, LA-Stab, and LA-Deg showed significant but reduced efficacy. 4OHT and LSZ102 were the least effective. *Figure 8* shows a structural analysis for RAL and 4OHT, which are representative of the most and least effective molecules in WT/Y537S *ESR1* MCF7 cells. In the most effective molecules (BZA, RAL, and RU39411), superposition of all monomers of WT and Y537S structures, based on alpha carbon positions, shows that H12 maintains a conserved AF-2 antagonist conformation that is well ordered in the AF2 cleft regardless of mutation (*Figure 8A/B*). In addition, a new hydrogen bond is observed between S537 and E380 (*Figure 8C*), which was previously computationally predicted for the Y537S-BZA structure (*Fanning et al., 2018b*). This hydrogen bond is not observed in monomers where crystal packing forms a hydrogen bond between D538 and a symmetry mate arginine, pulling 537 S away from E380 (*Figure 8—figure supplement 1*). Laso, LA-Stab, and LA-Deg showed intermediate efficacy. In these Y537S structures, H12 is slightly more dynamic and the S537-E380 hydrogen bond is no longer observed. 4OHT and LSZ102 were the least effective molecules. Both show significant Y537S H12 conformational changes, are significantly less ordered in the AF2 cleft, and no S537-E380 hydrogen bond is observed (*Figure 8D–F*).

Reduced Y537S efficacy correlates with decreased H12 surface area burial in the AF-2 Cleft. As H12 showed improved AF-2 cleft packing in the Y537S structures with the most effective molecules, we performed B-factor analysis to understand whether higher B-factors correlated with reduced anti-transcriptional efficacy. Indeed, the least effective molecules (4OHT and LSZ102) possess greater main-chain H12 B-factors (normalized to all main chain atoms) indicating a relatively poor packing in the AF-2 cleft compared to more effective molecules (*Figure 8—figure supplement 2*). We further calculated H12 buried surface area (BSA) Y537S LBD using proteins, interfaces, structures, and assemblies (*Krissinel and Henrick, 2005*) and found it correlates with transcriptional antagonism. BZA, RAL, and RU39411 show the greatest burial at 379.75, 377.87, and 325.21 Å (*Maximov et al., 2018*)

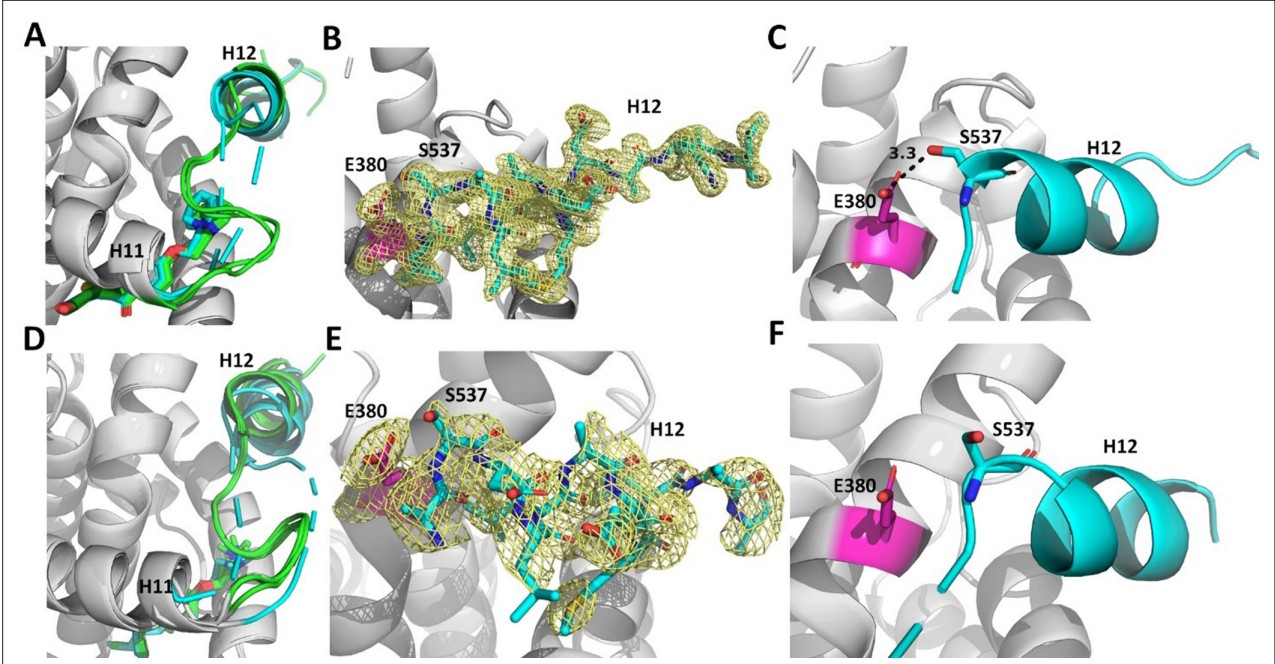

**Figure 8.** Enforcing helix 12 AF-2 cleft burial enhances Y537S estrogen receptor alpha (ERα) transcriptional inhibition. (**A**) Superposition of each monomer in the asymmetric unit of WT (green) or Y537S (cyan) ERα LBD in complex with RAL. (**B**) 2mFo-DFc difference map (yellow mesh) of the electron density around E380 (magenta) and H12 (cyan) of the Y537S-RAL structure contoured to 1.0 σ. (**C**) Hydrogen bond formed between E380 and S537 in Chain A of Y537S-RAL. (**D**) Superposition of each monomer in the asymmetric unit of WT (green) or Y537S (cyan) ERα in complex with 4OHT. (**E**) 2mFo-DFc difference map (yellow mesh) of the electron density around E380 (magenta) and H12 (cyan) of the Y537S-4OHT structure contoured to 1.0 σ. (**F**) Position of S537 relative to E380 in the Y537S-4OHT structure. Raloxifene PDBs: 7KBS and 7UJC and 4OHT PDBs: 5 W9C and 7UJ8.

The online version of this article includes the following source data and figure supplement(s) for figure 8:

**Figure supplement 1.** Representative crystal contact formed between D538 and R436 of a symmetry mate that pulls 537 S out of hydrogen bonding distance to E380.

**Figure supplement 2.** Selective estrogen receptor modulators (SERMs) and selective estrogen receptor degraders/downregulators (SERDs) with increased transcriptional antagonistic efficacies show improved helix 12 (**H12**) packing in the AF-2 cleft.

**Figure supplement 3.** 2mFo-DFc difference maps for antiestrogens in complex with WT and Y537S estrogen receptor alpha (ERα) LBD contoured to 1.5σ.

**Source data 1.** X-ray crystal structure data collection and refinement statistics.

respectively. Molecules with reduced potencies showed reduced BSA at 245.57, 56.13, 27.62, and 14.69 Å (*Maximov et al., 2018*) for LA-Stab, LSZ-102, LA-Deg, and 4OHT respectively. Together these structures show that SERMs, SERM/SERDs, or SERDs that favor H12 packing in the AF-2 cleft and promote an S537-E380 hydrogen bond and achieve the greatest anti-transcriptional efficacies in MCF7 breast cancer cells that harbor WT/Y537S ERα.

## Discussion

We show here how chemical manipulation of ERα H12 conformation and mobility can be used to tune receptor lifetime in live cells. SERMs extend ERα lifetime, increase receptor accumulation, and are antagonists in the breast and agonists in the bone and uterine epithelium (*Fanning and Greene, 2019*). Tissue-specific partial agonist activities stem from SERM inhibition of Activating Function-2 (AF-2) but not AF-1 genomic activities (*Shang and Brown, 2002*). SERDs decrease the quantity and lifetime of ERα in the cell and are pure antagonists by effectively prohibiting ERα transcriptional activities. Activating somatic mutations near ERα helix 12, Y537S and D538G, enable hormone therapy resistance in metastatic ER +breast cancers (*Toy et al., 2013*; *Jeselsohn et al., 2014*). Antiestrogens with improved potencies including bazedoxifene, fulvestrant, OP-1074 and GDC-0945 possess variable degrees of ERα degrading/downregulating activities but show significant potencies and efficacies

in mutant *ESR1* breast cancer cells (*Toy et al., 2013*; *Metcalfe et al., 2019*; *Fanning et al., 2018a*). Initial studies suggested that SERD activity was necessary for therapeutic potency (*Toy et al., 2013*); however, Lainé et al. showed that the SERM Laso induced marked anti-cancer activities in xenografts of Y537S *ESR1* breast cancer cells without fulvestrant-level ERα degradation (*Lainé et al., 2021*). In addition, *Wardell et al., 2020* concluded that antagonism rather than ERα degradation/downregulation drives most of fulvestrant efficacy in suppressing primary tumor growth in xenograft assays, based on observations that a dose of fulvestrant affording a non-significant reduction in ER levels led to significant growth suppression of a long-term estrogen-deprived breast cancer cell model over 4 weeks (). This is supported by observation in cell lines that fulvestrant retains transcriptional inhibition properties upon increases in ERα concentrations that lead to saturation of degradation. (*Lupien et al., 2007*) Based on this work, we examined the contribution of SERD activity to the potency and efficacy of transcriptional suppression by antiestrogens in breast cancer cells harboring Y537S or D538G *ESR1*. To understand whether chemical manipulation of ERα cellular lifetime correlates with improved antagonistic activities, we synthesized methylated derivatives of Laso that uniquely influence receptor cellular lifetime and accumulation. We compared these molecules to a comprehensive panel of SERMs, SERM/SERDs and SERDs for their effects on WT and mutant ERα cellular stability and transcriptional activity in breast cancer cells.

To quantify the influence of antiestrogens on ERα cellular lifetime, we developed a high-throughput system to observe WT, Y537S, or D538G ERα expression in living T47D breast cancer cells. Because induction of ERα modifications may take place even when degradation is absent, we also quantified differences in ERα SUMOylation, a post-translational modification that is induced by fulvestrant together with ubiquitination and that influences interaction with DNA and transcriptional activity in ER +MCF7 cells (*Traboulsi et al., 2019*). We discovered that addition of a 3R-methyl group on Laso's pyrrolidine induced both ERα SUMOylation and degrading activities, while addition of a 2S-methyl on the pyrrolidine leads to an accumulation of ERα in the breast cancer cells.

This approach also revealed an unexpected effect of each mutant on ERα expression and cellular lifetime. Most SERD molecules exhibited diminished capacities to affect Y537S ERα levels compared to WT. Two carboxylic acid-containing SERD molecules AZD9496 and LSZ102 unexpectedly switched to stabilizing Y537S, similar to 4OHT. Of note, AZD9496 did not induce SUMOylation with WT ERα, in keeping with a different conformation of the receptor observed in carboxylic acid-containing antiestrogens. Apart from this discrepancy, the general good degree of correlation between the impact of antiestrogens on SUMOylation of WT or mutant ER and their impact on receptor lifetime suggests either a cross-talk between SUMOylation and ubiquitination, possibly mediated by SUMO targeted ubiquitin ligases (STUbLs) such as RNF4, or similar structural determinants for addition of either marks by their respective enzymatic complexes. It will therefore be of interest in the future to explore whether carboxylic acid-containing antiestrogens selectively induce increased receptor ubiquitination, or lead to degradation by other mechanisms.

We were surprised that the Y537S mutant showed nearly identical unliganded/apo and E2-treated profiles, with enhanced cellular expression in the presence hormone to similar levels as WT in the absence of hormone. In addition to constitutive transcriptional activities and resistance to inhibition by tamoxifen, the Y537S also likely contributes to breast cancer pathology through increasing the quantity of active receptor in the cell. The D538G mutant was also surprising because it dampened the influence of SERDs on ERα SUMOylation, accumulation, and lifetime. It suppressed the stabilizing effects of SERMs on WT ERα, correlating with induction of weak SUMOylation of the D538G mutant but not WT receptor. The D538G *ESR1* mutant occurs with the greatest frequency in patients (*Chandarlapaty et al., 2016*) but its selective advantage has remained unclear. Our studies suggest that the diminished ability of cells to degrade D538G ERα coupled with its low-level constitutive transcriptional activities contributes a pathological advantage within the tumor in the presence of first-line endocrine therapies. Further studies will reveal whether these alterations to ERα lifetime influence breast cancer pathological endpoints and therapeutic response.

Transcriptional reporter gene assays in MCF7 cells harboring WT, WT/Y537S, or WT/D538G *ESR1* were used to understand whether the degrees of ERα degradation and/or SUMOylation correlated with transcriptional inhibitory activities. We identified antiestrogens across the SERM-SERD spectrum that demonstrated significantly improved transcriptional repressive activity over 4OHT in these cells. Interestingly, RU39411, which significantly enriched ERα levels (SERM-like), showed enhanced

transcriptional inhibition compared to most molecules. Despite their unique influences on WT ERα cellular stability, the methylated Laso derivatives showed identical transcriptional antagonistic efficacies in the WT and mutant cell lines compared to unmodified Laso in our assay.

Alkaline phosphatase activities in Ishikawa uterine cell lines were used to understand whether observed SERM or SERD influences on ERα degradation correlated with receptor-dependent uterotrophic activities. Here, the SERM-like LA-Stab showed a robust induction of AP activity, but not to the extent of 4OHT or Z-endoxifen. Laso showed a modest induction and LA-Deg showed no induction of AP activity. Surprisingly, 4OHT was more uterotrophic than Z-endoxifen, another tamoxifen metabolite. These results suggest that Z-endoxifen is less SERM-like that 4OHT and may explain its improved efficacy in letrazole-resistant models of ER +breast cancer (*Jayaraman et al., 2020*).

Cellular proliferation assays in MCF7 cells with WT/Y537S *ESR1* were used to reveal the role of ERα degradation in anti-proliferative activities. We showed that 4OHT, fulvestrant, Laso, LA-Deg, and LA-Stab can reduce E2-stimulated proliferation. Fulvestrant and Laso show improved efficacies at lower concentrations, while LA-Stab and LA-Deg did not show improved anti-proliferative activities. However, no molecule completely repressed the proliferation of these cells. Our BRET, transcription, and proliferation data are consistent with our observations that SERMs have similar efficacies at suppressing cofactor recruitment in WT and $ESR1_{mut}$ cells. Together, these data suggest that SERM or SERD influences on ERα cellular lifetime may not directly correlate with transcriptional antagonistic and anti-proliferative activities of *ESR1* mutant breast cancer cells. Rather, engineered chemical reduction of ERα cellular accumulation is critical for tuning tissue- and gene-specific agonistic/ partial agonistic activities (*Fanning and Greene, 2019*) and preventing AF-1 dependent mechanisms of endocrine resistance (*Massarweh et al., 2008*).

X-ray crystal structures of WT and Y537S ERα LBD in complex with SERMs, SERM/SERDs, and a SERD showed that the most efficacious molecules enforced H12 burial within the AF-2 cleft. We previously showed that BZA resisted the impact of the Y537S mutation on its antagonist H12 conformation, which was perturbed for the less effective 4OHT (*Fanning et al., 2018b*). Of the structures, RU39411 (SERM), RAL (SERM/SERD), and BZA (SERM/SERD) demonstrated the greatest transcriptional inhibitory efficacies in WT/Y537S *ESR1* MCF7 cells. These molecules engaged a new S537-E380 hydrogen bond that enforced a well-ordered antagonistic H12 conformation that corresponded to significant surface area burial within the AF-2 cleft. Conversely, molecules with diminished Y537S transcriptional potencies poorly enforced this conformation as H12 showed variable conformations and reduced surface area burial in the AF-2 cleft.

Overall, our studies show how minor chemical modifications lead to switches in ERα stabilizing or degrading activities onto an antiestrogen scaffold, likely via modulation of ERα post-translational modification. While these activities may not confer an antagonistic benefit in breast cancer cells, they are important drivers of AF-1 mediated tissue-specific activities (*Shang and Brown, 2002*). ERα proteolysis is critical to normal activity and E2 binding induces phosphorylation at position Y537, which recruits E3 ubiquitin ligases (*Sun et al., 2012*). Therefore, further understanding of the role played by Y537 phosphorylation in ERα SERM or SERD antagonism in breast cancer cells requires further study. Importantly, we showed that SERMs and SERDs differentially affect Y537S H12 binding in the AF-2 cleft. SERDs, like fulvestrant, increase H12 conformational mobility (*Fanning et al., 2018b*; *Pike et al., 2001*) and enhance binding of corepressor proteins such as NCOR (*Webb et al., 2003*) that are important contributors to anti-cancer activities. Therefore, it is likely that SERMs and SERDs differentially affect the repertoire of coregulator proteins that associate with Y537S ERα and contribute to anti-cancer activities. Revealing these changes will be critical for developing new competitive antiestrogens that maintain activities in the presence of Y537S and D538G *ESR1* mutations.

## Materials and methods
### Cell culture
MCF-7 and T47D cells were purchased directly from ATCC. MCF7 WT/Y537S cells were generously donated by Dr Ben Ho Park. MCF-7 cells were cultured in DMEM supplemented with 10% FBS and 1% penicillin-streptomycin. T47D cells were cultured in RPMI supplemented with 10% FBS and 1% penicillin-streptomycin. Cells were STR profiled to confirm identify and routinely subjected to mycoplasma testing.

## Live cell assay of halo-ER degradation

T47D cells with stable TetON HT WT, Y537S, or D538G ERα were plated in a 6 well dish at a density of 15,000 and 30,000 cells, respectively. These cells were cultured for 48 hr in phenol-free media supplemented with charcoal-stripped FBS in puromycin. Following this incubation, 1 µg/mL dox and 1 µM G618 were added to each well. Cells were then treated with increasing concentrations of our compounds of interest (0.00512, 0.0256, 0.128, 0.64, 3.2, 16, 80, 400, 2000, and 5000 nM) for 24 hr. After treatment, the cells were imaged using an Incucyte S3. ER degradation was quantified by using the red channel integrated intensity per image normalized to the phase channel confluence area. Assays were performed twice with three technical replicates each.

## Chemical synthesis

### 1-[4-(2-Benzyloxy)ethoxy]phenyl-2-bromo-6-methoxy-3,4-dihydronaphthalene (5)

was prepared starting from 4-bromophenol in four steps according to a literature procedure with slightly modification (*Simpson et al., 1987*).

*Figure 9* shows the synthetic route used to generate the Laso analogs. A mixture of 4-bromophenol (40.0 g, 231 mmol), ethylene carbonate (38.0 g, 432 mmol) and $K_2CO_3$ (68.0 g, 492 mmol) in DMF (200 mL) was heated for 36 hr at 100 °C and then cooled. The reaction mixture was poured into water (500 mL), extracted with ether (3 × 400 mL) and the combined organic layers was washed with 5% NaOH (250 mL), brine and dried over $Na_2SO_4$. The solid was filtered off and the solvent was removed by rotary evaporator under reduced pressure to give 2-(4-bromophenoxy)ethanol (**2**) as colorless solid (49.5 g, 99% yield). [1]H NMR ($CDCl_3$/TMS) δ7.37 (m, 2 H), 6.79 (m, 2 H), 4.04 (m, 2 H), 3.96 (m, 2 H), 2.14 (t, 1 H, J = 6.0 Hz); [13]C NMR ($CDCl_3$) δ157.9, 132.5, 116.5, 113.4, 69.5, 61.5 ppm.

Under argon to the solution of **2** (48.8 g, 225 mmol) in dry DMF (650 mL), NaH (95%, 8.53 g, 338 mmol) was added at 0 °C. After stirred at rt for 30 min, tetra-*n*-butylammonium iodide (872 mg, 2.36 mmol) and benzyl chloride (29.9 g, 236 mmol) was added. The reaction mixture was stirred at rt overnight and then quenched with saturated aqueous ammonium chloride (150 mL). The product was extracted with dichloromethane (4 × 100 mL). The combined organic layers were washed with brine and dried over $Na_2SO_4$. The solvent was removed and the residue was purified by silica gel chromatography, eluting with 10% ethyl acetate to give 4-[2-(benzyloxy) ethoxy] bromobenzene (**3**) (57.7 g, 74% yield). [1]H NMR ($CDCl_3$/TMS) δ7.37–7.28 (m, 7 H), 6.81 (m,2H), 4.63 (s, 2 H), 4.11 (m, 2 H), 3.82 (m, 2 H); [13]C NMR ($CDCl_3$) δ158.1, 138.1, 132.4, 128.6, 127.9, 116.6, 113.2, 73.6, 68.5, 67.8 ppm.

Under argon a mixture of **3** (28 g, 91.2 mmol) and magnesium turnings (2.32 g, 95.8 mmol) in THF (300 mL) was heated under reflux for 24 h with the addition of several iodine particles to initiate the reaction. The solution was cooled to rt and 6-methoxy-3,4-dihydronaphthalen-1(2 H)-one (11.7 g, 66.4 mmol) in THF (100 mL) was added dropwise. After the reaction mixture was stirred at rt for 20 hr, the reaction was quenched with 5% HCl (100 mL). The product was extracted with ether, dried over $Na_2SO_4$. The solvent was removed, the residue was isolated by silica gel chromatography, eluting with 10% ethyl acetate in hexane to give 1-[4-(2-benzyloxy)ethoxy]phenyl-6-methoxy-3,4-dihydronaphthalene (**4**) as light-yellow oil (13.9 g, 54% yield). [1]H NMR ($CDCl_3$/TMS) δ7.40–7.20 (m, 7 H), 7.00–6.90 (m, 3 H), 6.76 (d, 1 H, J = 2.4 Hz), 6.63 (dd, 1 H, J = 8.6, 2.6 Hz), 5.91 (t, 1 H, J = 4.4 Hz), 4.65 (s, 2 H), 4.18 (m, 2 H), 3.85 (m, 2 H), 3.79 (s, 3 H), 2.81 (t, 2 H, J = 8.0 Hz), 2.36 (m, 2 H); [13]C NMR ($CDCl_3$) δ158.6, 158.1, 139.0, 138.8, 138.2, 133.7, 129.8, 128.6, 127.92, 127.85, 126.7, 124.6, 114.5, 113.9, 110.8, 73.5, 68.7, 67.6, 55.4, 29.0, 23.6 ppm.

Bromine (2.0 mL, 39.5 mmol) was added slowly to the solution of **4** (13.87 g, 35.9 mmol) in THF (120 mL) at 0 °C. After 5 min, triethylamine (5.5 mL, 39.5 mmol) was added while the mixture was stirred vigorously. The reaction mixture was stirred at rt for 10 min. The reaction mixture was washed with 5% $Na_2S_2O_3$, extracted with ether, washed with brine and dried over $Na_2SO_4$. The solvent was removed and the residue was isolated by silica gel chromatography, eluting with 10% ethyl acetate in hexane to give 1-[4-(2-benzyloxy)ethoxy]phenyl-2-bromo-6-methoxy-3,4-dihydronaphthalene (**5**) as brown solid (15.6 g, 94% yield). [1]H NMR ($CDCl_3$/TMS) δ7.40–7.24 (m, 5 H), 7.15–7.10 (m, 2 H), 7.00–6.94 (m, 2 H), 6.70 (d, 1 H, J = 2.4 Hz), 6.60–6.50 (m, 2 H), 4.66 (s, 2 H), 4.20 (m, 2 H), 3.87 (m, 2 H), 3.77 (s, 3 H), 2.95 (m, 4 H); [13]C NMR ($CDCl_3$) δ158.8, 158.1, 138.2, 137.6, 136.2, 132.4, 131.0, 129.4, 128.6, 127.92, 127.85, 127.5, 120.7, 114.5, 113.7, 110.0, 73.5, 68.7, 67.5, 55.4, 35.2, 30.1 ppm.

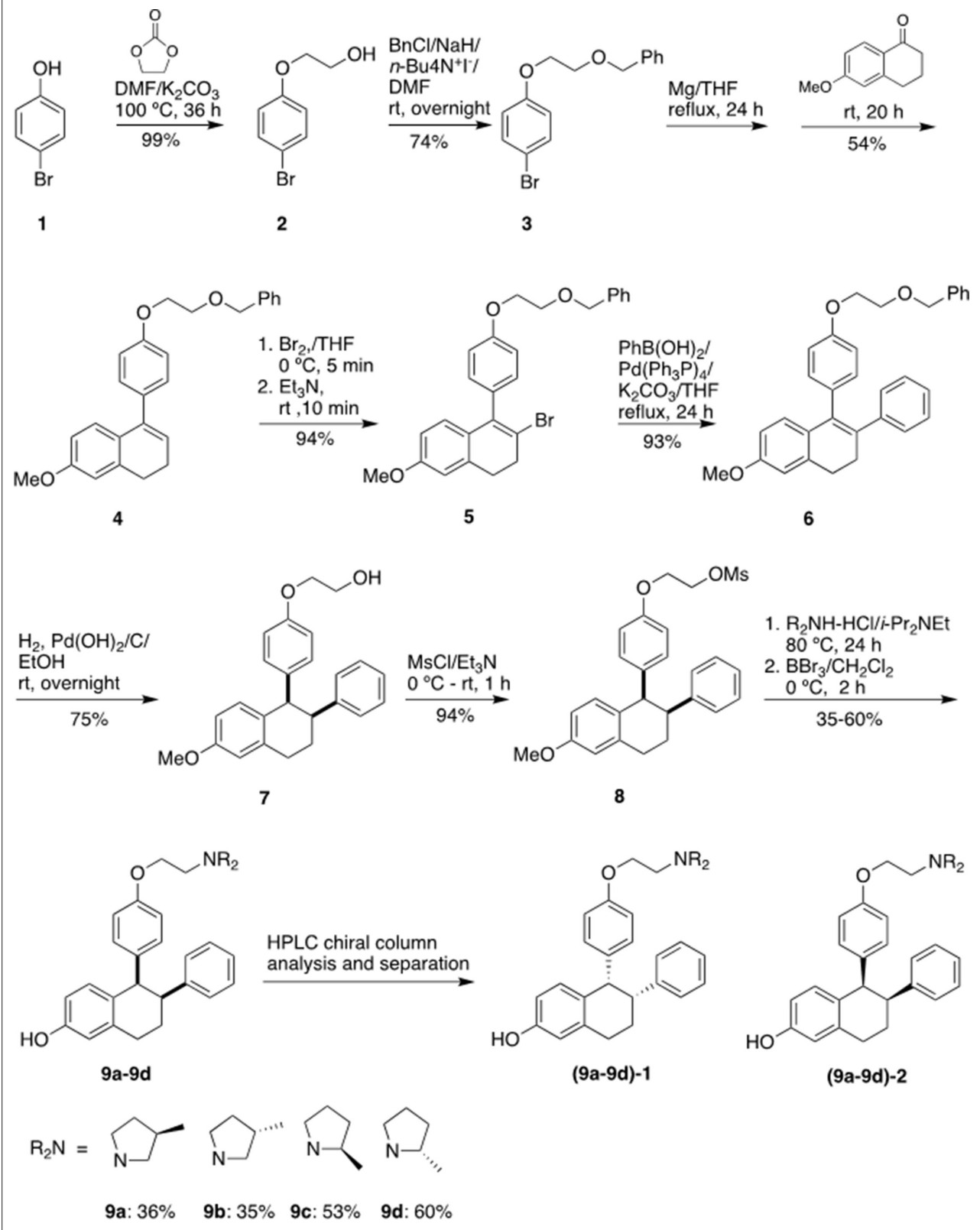

**Figure 9.** Experimental for the synthesis of lasofoxifene analogues (**9a-9d**).

## 1-[4-(2-Benzyloxy) ethoxy] phenyl-6-methoxy-2-phenyl-3,4-dihydronaphthalene (6)

Under argon a mixture of **4** (3.84 g, 8.25 mmol), phenylboronic acid (1.11 g, 9.1 mmol), $Pd(PPh_3)_4$ (286 mg, 0.247 mmol) and potassium carbonate (2.28 g, 16.5 mmol) in dry THF (100 mL) was stirred at reflux for 24 h. After the mixture was cooled down, water (30 mL) was added. The organic layer was separated. The aqueous layer was extracted with ether, washed with brine. The organic layers were combined and dried over $Na_2SO_4$. The solvent was removed, the residue was isolated by silica gel chromatography, eluting with 10% ethyl acetate in hexane to give **6** (3.55 g, 93% yield). $^1$H NMR ($CDCl_3$/TMS) δ7.40–7.20 (m, 6 H), 7.15–6.90 (m, 6 H), 6.82–6.78 (m, 3 H), 6.74 (d, 1 H, *J* = 8.8 Hz), 6.60 (dd, 1 H, *J* = 8.8, 2.8 Hz), 4.66 (s, 2 H), 4.13 (m, 2 H), 3.83 (m, 2 H), 3.82 (s, 3 H), 2.95 (m, 2 H), 2.79 (m, 2 H); $^{13}$C NMR ($CDCl_3$) δ158.5, 157.5, 143.4, 138.2, 137.8, 134.9, 134.4, 132.4, 132.2, 130.6, 128.5, 128.4, 127.9, 127.8, 127.7, 127.6, 125.8, 114.2, 113.3, 110.9, 73.5, 68.6, 67.4, 55.4, 30.9, 29.1 ppm.

## 1-[4-(2-Hydroxy) ethoxy] phenyl-6-methoxy-2-phenyl-1,2,3,4-tetrahydronaphthalene (7)

To the solution of **6** (3.47 g, 7.5 mmol) in a mixed solvent of ethanol/ethyl acetate (2/1, v/v, 150 mL), palladium hydroxide on carbon (20% wt., 263 mg, 0.375 mmol) was added. After exchange the air with argon, the mixture was then stirred under a hydrogen balloon overnight. TLC showed no starting material left. The catalyst was filtered off. The solvent was removed and the residue was isolated by silica gel chromatography, eluting with 20% ethyl acetate in hexane to give **7** as a white foam (2.105 g, 75% yield). $^1$H NMR ($CDCl_3$/TMS) δ7.15–7.05 (m, 3 H), 6.85–6.70 (m, 4 H), 6.63 (dd, 1 H, *J* = 8.4, 2.8 Hz), 6.49 (d, 2 H, *J* = 8.8 Hz), 6.30 (d, 2 H, *J* = 8.4 Hz), 4,21 (d, 1 H, *J* = 6.0 Hz), 3.83 (m, 2 H), 3.79 (m, 2 H), 3.73 (s, 3 H), 3.32 (m, 1 H), 3.02 (m, 2 H), 2.89 (*brs*, 1 H), 2.14 (m, 1 H), 1.78 (m, 1 H); $^{13}$C NMR ($CDCl_3$) δ157.7, 156.6, 144.2, 137.6, 134.9, 132.1, 131.4, 131.3, 128.1, 127.6, 125.9, 112.84, 112.82, 112.5, 68.9, 61.1, 55.0, 50.1, 45.3, 30.0, 21.8 ppm. HRMS calcd for $C_{25}H_{30}NO_3$ [$MNH_4^+$] 392.2226, found 392.2222.

## 1-[4-(2-Mesoxy) ethoxy] phenyl-6-methoxy-2-phenyl-1,2,3,4-tetrahydronaphthalene (8)

Triethylamine (0.63 mL, 4.52 mmol) was added to a solution of **7** (848 mg, 2.26 mmol) and MsCl (0.35 mL, 4.52 mmol) in THF (25 mL) at 0 °C. The reaction mixture was stirred at rt for 1 hr. The reaction was quenched with water, extracted with ethyl acetate, and dried over MgSO4. The solvent was removed and the residue was purified by silica gel chromatography, eluting with 20% ethyl acetate in hexane to give the mesylate **8** (0.90 g, 94% yield). $^1$H NMR ($CDCl_3$/TMS) δ7.18–7.10 (m, 3 H), 6.85–6.70 (m, 4 H), 6.66 (dd, 1 H, *J* = 8.4, 2.4 Hz), 6.50 (d, 2 H, *J* = 8.4 Hz), 6.33 (d, 2 H, *J* = 8.4 Hz), 4,44 (m, 2 H), 4.23 (d, 1 H, *J* = 4.8 Hz), 4.05 (m, 2 H), 3.78 (s, 3 H), 3.36 (m, 1 H), 3.06 (m, 2 H), 2.99 (s, 3 H), 2.16 (m, 1 H), 1.82 (m, 1 H); $^{13}$C NMR ($CDCl_3$) δ157.9, 156.0, 144.2, 137.8, 135.7, 132.0, 131.50, 131.45, 128.1, 127.8, 126.0, 113.0, 112.9, 112.6, 68.3, 65.6, 55.2, 50.2, 45.3, 37.6, 30.1, 21.9 ppm. HRMS calcd for $C_{26}H_{28}O_5SNa$ [$MNa^+$] 475.1555, found 475.1550.

### Typical procedure for the synthesis of 9a-9d

The synthesis of **9** a is representative: To the solution of **8** (85 mg, 0.20 mmol) in anhydrous DMF (5 mL), (3 R)–3-methylpyrrolidine-HCl salt (242 mg, 2.0 mmol) and *i*-Pr$_2$NEt (700 uL, 4.0 mmol) were added. After the mixture was stirred at 80 °C for 24 h, TLC showed the reaction was completed. The solvent was removed, and the residue was dissolved into ether, washed with saturated NaHCO$_3$, brine, and dried over Na$_2$SO$_4$. After filtration, the solvent was removed, the residue was dried under vacuum overnight, and then redissolved into dry dichloromethane (10 mL). Under argon the dichloromethane solution was cooled to –78 °C. BBr$_3$ (1.0 M in CH$_2$Cl$_2$, 2.0 mL, 2.0 mmol) was added. After the reaction mixture was warmed and stirred at 0 °C for 2 h, the reaction was quenched with Na$_2$S$_2$O$_3$ solution at 0 °C. The mixture was basified with 2 N NaOH. extracted with CH$_2$Cl$_2$, washed with brine and dried over MgSO$_4$. The solvent was removed, the residue was purified by silica gel chromatography, eluting with 4% methanol in dichloromethane containing 1% triethylamine to give product **9** a (31 mg, 36% yield). (***Burstein et al., 2019***) H NMR ($CDCl_3$/TMS) δ7.15 (m, 3 H), 6.80–6.62 (m, 4 H), 6.58 (dd, 1 H, *J* = 8.4, 2.4 Hz), 6.42 (m, 2 H), 6.26 (m, 2 H), 4.19 (d, 1 H, *J* = 5.2 Hz), 3.98 (m, 2 H), 3.35 (m, 1 H), 3.15–2.55

(m, 7 H), 2.45–2.25 (m, 1 H), 2.25–2.05 (m, 3 H), 1.76 (m, 1 H), 1.42 (m, 1 H), 1.05 (d, 3 H, $J$ = 6.8 Hz); $^{13}$C NMR (CDCl$_3$) δ156.7, 155.4, 144.6, 137.6, 135.0, 131.5, 131.3, 130.8, 128.30, 127.8, 126.0, 115.15, 115.12, 114.2, 112.8, 65.8, 62.5, 62.3, 55.5, 54.6, 54.4, 50.3, 45.5, 32.43, 32.39, 31.9, 31.8, 30.0, 22.0, 19.9 ppm; HRMS calcd for C$_{29}$H$_{34}$NO$_2$ [MH$^+$] 428.2590 found 428.2590. HPLC analysis and purification with Chiral pack IBN-3 column, 0.75 mL/min, mobile phase: 0.1% Et$_2$NH in MeOH. UV 279 nm, **9** a-1, retention time 8.05 min, **9** a-2, retention time 9.62 min.

9b (30 mg, 35% yield) was prepared from **8** (85 mg, 0.20 mmol), (3 S)–3-methylpyrrolidine-HCl salt (121 mg, 1.0 mmol) and $i$-Pr$_2$NEt (348 µL, 2.0 mmol) according to the typical procedure: (*Burstein et al., 2019*) H NMR (CDCl$_3$/TMS) δ7.14 (m, 3 H), 6.80–6.62 (m, 4 H), 6.55 (dd, 1 H, $J$ = 8.4, 2.8 Hz), 6.37 (d, 2 H, $J$ = 8.8 Hz), 6.26 (d, 2 H, $J$ = 7.6 Hz), 4.19 (d, 1 H, $J$ = 4.8 Hz), 3.96 (m, 2 H), 3.35 (m, 1 H), 3.15–2.75 (m, 6 H), 2.60 (m, 1 H), 2.40–2.25 (m, 1 H), 2.20–2.00 (m, 3 H), 1.75 (m, 1 H), 1.40 (m, 1 H), 1.05 (d, 3 H, $J$ = 6.8 Hz); $^{13}$C NMR (CDCl$_3$) δ156.8, 155.5, 144.7, 137.6, 134.9, 131.6, 131.4, 130.7, 128.4, 127.8, 126.0, 115.23, 115.18, 114.3, 112.8, 65.9, 62.6, 62.4, 55.63, 55.60, 54.6, 54.4, 50.3, 45.6, 32.51, 32.45, 31.9, 31.8, 30.1, 22.0, 20.1, 20.0 ppm; HRMS calcd for C$_{29}$H$_{34}$NO$_2$ [MH$^+$] 428.2590 found 428.2589. HPLC analysis with Chiral pack IBN-3 column, 0.75 mL/min, mobile phase: 0.1% Et$_2$NH in EtOH. UV 279 nm, **9b-1**, retention time 5.87 min, **9b-2**, retention time 6.67 min.

**9** c (52 mg, 53% yield) was prepared from **8** (100 mg, 0.23 mmol), (2 R)–2-methylpyrrolidine-HCl salt (280 mg, 2.3 mmol) and $i$-Pr$_2$NEt (800 uL, 4.6 mmol) according to the typical procedure: $^1$H NMR (CDCl$_3$/TMS) δ7.15 (m, 3 H), 6.80–6.62 (m, 4 H), 6.55 (m, 1 H), 6.46 (d, 1 H, $J$ = 8.4 Hz), 6.41 (d, 1 H, $J$ = 8.8 Hz), 6.27 (m, 2 H), 4.19 (d, 1 H, $J$ = 4.8 Hz), 3.98 (m, 2 H), 3.35–3.15 (m, 3 H), 3.02–2.90 (m, 2 H), 2.50–2.40 (m, 2 H), 2.29 (m,1H), 2.20–2.00 (m, 1 H), 2.00–1.90 (m, 1 H), 1.90–1.60 (m, 3 H), 1.55–1.40 (m, 1 H), 1.19 (m, 3 H); $^{13}$C NMR (CDCl$_3$) δ156.8, 155.3, 155.2, 144.63, 144.60, 137.7, 135.0, 134.9, 131.6, 131.5, 131.4, 131.0, 130.8, 128.32, 128.29, 127.8, 126.0, 115.3, 115.2, 114.4, 114.2, 112.9, 112.8, 66.4, 66.2, 61.1, 60.9, 54.5, 54.4, 53.0, 50.3, 45.6, 32.33, 32.26, 30.1, 22.0, 21.8, 21.7, 18.5, 18.4 ppm.; HRMS calcd for C$_{29}$H$_{34}$NO$_2$ [MH$^+$] 428.2590 found 428.2591. HPLC analysis with Chiral pack IBN-3 column, 0.75 mL/min, mobile phase: 0.1% Et$_2$NH in EtOH. UV 279 nm, **9** c-1, retention time 5.88 min, **9** c-2, retention time 6.57 min.

**9d** (59 mg, 60% yield) was prepared from **8** (100 mg, 0.23 mmol), (2 S)–2-methylpyrrolidine-HCl salt (280 mg, 2.3 mmol) and $i$-Pr$_2$NEt (800 µL, 4.6 mmol) according to the typical procedure: (*Burstein et al., 2019*) H NMR (CDCl$_3$/TMS) δ7.15 (m, 3 H), 6.80–6.62 (m, 4 H), 6.55 (m, 1 H), 6.47 (d, 1 H, $J$ = 8.8 Hz), 6.42 (d, 1 H, $J$ = 8.8 Hz), 6.28 (m, 2 H), 4.19 (d, 1 H, $J$ = 4.8 Hz), 3.98 (m, 2 H), 3.40–3.15 (m, 3 H), 3.02–2.90 (m, 2 H), 2.50–2.40 (m, 2 H), 2.26 (m, 1H), 2.20–2.15 (m, 1 H), 2.00–1.87 (m, 1 H), 1.87–1.60 (m, 3 H), 1.55–1.40 (m, 1 H), 1.16 (m, 3 H); $^{13}$C NMR (CDCl$_3$) δ156.8, 155.31, 155.26, 144.62, 144.60, 137.71, 137.69, 135.0, 134.9, 131.6, 131.5, 131.3, 130.9, 130.8, 128.30, 128.28, 127.8, 126.0, 115.3, 115.2, 114.4, 114.3, 112.9, 112.8, 66.5, 66.3, 61.0, 60.8, 54.6, 54.4, 53.0, 50.3, 45.6, 32.34, 32.28, 30.1, 22.0, 21.8, 21.7, 18.6, 18.4 ppm; HRMS calcd for C$_{29}$H$_{34}$NO$_2$ [MH$^+$] 428.2590 found 428.2591. HPLC analysis and purification with Chiral pack IBN-3 column, 0.75 mL/min, mobile phase: 0.1% Et$_2$NH in MeOH. UV 279 nm, **9d-1**, retention time 7.94 min, **9d-2**, retention time 9.11 min.

## BRET assays

HEK293 HTS cells were maintained at 37 degree celsius in Dulbecco's modified Eagle's medium (DMEM) (Wisent) supplemented with 10% fetal bovine serum (FBS) (Sigma), 2% L-glutamine (Wisent) and 1% penicillin-streptomycin (Wisent). Two days prior to transfection, HEK293 HTS cells were washed twice in PBS and seeded in 15 cm petri dishes at a density of 5×10$^6$ in phenol red-free DMEM supplemented with 1charcoal-stripped FBS (Sigma) and 1% penicillin-streptomycin. Forty-eight hr later, cells were co-transfected with expression plasmids for WT, Y537S, or D538G ERα fused to RlucII and YFP-SUMO3 (*Hilmi et al., 2012*; *Cotnoir-White et al., 2018*) or SRC1/NCOA1 RID-Topaz YFP (*Liao et al., 2002*). Transfections were performed using linear polyethylenimine (Polysciences Inc) at a ratio of 3 µg of PEI to 1 µg of DNA per 1.25×10$^6$ cells and aliquoted at 125,000 cells per well in white 96 well assay plates (Corning). BRET assays were carried out 48 h post-transfection by replacing cell media by 1 x HBSS (Wisent) supplemented with 4.5 g/L dextrose and either compounds or vehicle and incubating for 2–3 h at 37 degree celsius. For dose-response experiments, compounds were serially diluted 1:3 in 1 x HBSS from a maximum concentration of 9 µM. Coelenterazine H (Nanolight Technologies) was added to a final concentration of 5 µM and readings were taken following a 5 min incubation at room temperature using a Mithras LB 940 microplate reader (Berthold Technologies).

Net BRET ratios were calculated as described previously (*Hilmi et al., 2012*). All experiments were performed with three to five biological replicates comprised of three technical replicates each. Best-fit $EC_{50}$ and BRETMax values were determined with Prism. Curves represent average values +/-SEM.

## Live cell assay of ERE transcriptional response

MCF-7 cells with WT, WT/Y537S, and WT/D538G with a 3×-ERE-GFP reporter gene construct with a CMV promoter were plated in a 6 well dish at a density of 15,000 and 30,000 cells, respectively. Cells were cultured for 48 hr in media supplemented with charcoal-stripped FBS. Subsequently, cells were treated with increasing concentrations of compounds of interest (0.00512, 0.0256, 0.128, 0.64, 3.2, 16, 80, 400, 2000, and 5000 nM) for 48 hr. After treatment, the cells were imaged using an Incucyte S3. ERE transcriptional response was quantified by using the green channel integrated intensity per image normalized to the phase channel confluence area. Assays were performed twice with three technical replicates each.

## Alkaline phosphatase activity

This assay was performed with Ishikawa cells exactly as previously described (*Fanning et al., 2018a*). Briefly, approximately 15,000 Ishikawa cells were plated onto a 96-well plate in serum starved media. After 4 hr, cells were treated with vehicle, hormone, SERM, or SERD. Cells were incubated for 72 hr, media removed, and frozen at –80 °C for at least 12 hr. Thawed plates were incubated with p-nitrophenyl phosphate (NEB, #P0757L) for 1–3 hr at 37 °C. Absorbance was read at 405 nm on a BioTek Cytation 5 plate reader. These assays were performed three independent times with three technical replicates each.

## Cellular proliferation assay

A BioTek Cytation 5 with BioSpa was used for automatic cell counting experiments. MCF7 WT/Y537S cells were plated in 96-well plates at 750 cells per well in phenol-free medium. After 24 hr, cells placed in charcoal-stripped FBS and allowed to acclimate for 48 hr. Subsequently, cells were treated with 1, 50, and 1000 nM antiestrogen and 1 nM E2 then placed in the BioSpa. Cells were automatically imaged by bright-field and phase contrast every 6 hr for a total of 84 hr. Cell counts were analyzed using the label-free cell counting protocol on BioTek Gen5 software. These assays were performed three independent times with three technical replicates each.

## Protein expression and purification

A gene containing a hexa-His-TEV fusion of the ERα LBD, residues 300–550 with C381S, C417S, C530S, and L536S in pET21(a)+was used for all WT ERα LBD x-ray crystal structures, as this construct enables the adoption of a canonical antagonist conformation of the receptor (*Fanning et al., 2018a*). An identical construct with but with an intact 536 L and Y537S was generated using Q5 site-directed mutagenesis (Promega) and was used for all Y537S x-ray crystal structure determinations. *E. coli* BL21(DE3) were used for all recombinant protein expression. An overnight culture at 37 °C was inoculated with a single colony in 50 mL LB broth containing 100 µg/mL ampicillin. 5 mL of the overnight culture was used to inoculated each of 10 L of LB-ampicillin, which were allowed to grow at 37 °C with shaking until an $OD_{600}$ = 0.6 was reached, indicating log-phase growth. Protein expression commenced upon the addition of 0.3 mM IPTG and continued overnight at 16 °C. Cells were harvested by centrifugation at 4000 *g* for 15 min. Cells were resuspended in 5 mL/g cell paste in 50 mM HEPES pH 8.0, 250 mM NaCl, 20 mM imidazole pH 8.0, 0.5 mM TCEP, and 5% glycerol with the addition of 1 EDTA-free cOmplete protease inhibitor cocktail tablet per 50 mL lysate (Roche). Cells were lysed by sonication on ice with stirring and the lysates were clarified by centrifugation at 20,000 *g* for 30 min at 4 °C. The clarified lysate was loaded onto 2.5 mL of pre-equilibrated Ni-NTA resin and washed with 5 column volumes of 50 mM HEPES pH 8.0, 250 mM NaCl, 20 mM imidazole pH 8.0, 0.5 mM TCEP, and 5% glycerol. Protein was eluted with 50 mM HEPES pH 8.0, 250 mM NaCl, 400 mM imidazole pH 8.0, 0.5 mM TCEP, and 5% glycerol. A 1:200 mol:mol ratio of hexa-His-TEV protease was added to the eluent and the solution was dialyzed in 4 L of 50 mM HEPES pH 8.0, 250 mM NaCl, 20 mM imidazole pH 8.0, 0.5 mM TCEP, and 5% glycerol overnight at 4° with stirring. ERα was separated from the hexa-His tag and His-TEV protease by passing it over pre-equilibrated Ni-NTA resin and collecting the flow-through. The protein was concentrated to 5 mL, but no higher than 15 mg/mL and placed over

a Superdex 200 HiLoad 200 16/600 size exclusion column that was equilibrated with 25 mM HEPES pH 8.0, 150 mM NaCl, 0.5 mM TCEP, and 5% glycerol. A small precipitate peak was observed at the leading edge of the column followed by a well-resolved single peak corresponding to the correct molecular weight of approximately 30,000 Da for ERα LBD. Fractions corresponding to these proteins were concentrated to ≥ 10 mg/mL, flash frozen, and stored at –80 °C for later use.

### X-ray crystal structure determination

Purified LBD was incubated with 2 mM of each compound between 4 and 16 hr at 4 °C prior to crystal screens. The mixture was centrifuged at 20,000 $g$ for 30 min at 4 °C to remove precipitated ligand/protein. Hanging drop vapor diffusion was used to generate all crystals. 2 μL protein at 5–15 mg/mL was mixed with 2 μL mother liquor using Hampton VDX plates (Hampton Research, HR3-140). All crystals were grown in 5%–20% PEG 3350 or PEG 8000, pH 6–8.0, 200 mM MgCl$_2$. For each complex, clear crystals grew at room temperature. Crystals emerged between 16 hr and 2 weeks. For BZA, RAL, 4-OHT, Laso, 48–2, and LSZ102 complexes these crystals appeared as hexagonal pucks. For RU39411 and 49–2 complexes crystals were rhombohedral. Crystals were either directly frozen in mother liquor or incubated in mother liquor with the addition of 25% glycerol as cryo-protectants. All x-ray data sets were collected at the Advanced Photon Source, Argonne National Laboratories, Argonne, Illinois on the SBC 19-BM beamline (0.97 Å). *Figure 8—figure supplement 3* shows representative 2mFo-DFc difference maps for ligands in the ligand binding pocket. *Figure 8—source data 1* contains data collection and refinement statistics.

## Acknowledgements

Funding from Susan G Komen Foundation CCR19608597 (SWF), Ludwig Fund for Metastasis Research (GLG), and Canadian Institutes of Health Research (SM). Results shown in this report are derived from work performed at Argonne National Laboratory (ANL), Structural Biology Center (SBC) at the Advanced Photon Source (APS), under U.S. Department of Energy, Office of Biological and Environmental Research contract DE-AC02-06CH11257.

---

## Additional information

### Competing interests

Sean W Fanning: In the interest of transparency, Dr. Fanning's laboratory receives sponsored research funds from Olema Oncology Inc. Olema was not involved in this study. This work has no impact on the company. The other authors declare that no competing interests exist.

### Funding

| Funder | Grant reference number | Author |
| --- | --- | --- |
| Susan G. Komen | CCR19608597 | Sean W Fanning |
| Ludwig Fund for Metastasis Research | | Geoffrey L Greene |
| Canadian Institutes of Health Research | | Sylvie C Mader |

The funders had no role in study design, data collection and interpretation, or the decision to submit the work for publication.

### Author contributions

David J Hosfield, Conceptualization, Data curation, Formal analysis, Investigation, Methodology, Project administration, Supervision, Visualization, Writing – original draft, Writing – review and editing; Sandra Weber, Data curation, Formal analysis, Investigation, Visualization, Writing – original draft, Writing – review and editing; Nan-Sheng Li, Formal analysis, Methodology; Madline Sauvage, Data curation, Formal analysis, Investigation, Methodology, Visualization; Carstyn F Joiner, Data curation, Methodology; Govinda R Hancock, Data curation, Visualization, Writing – review and editing; Emily A

Sullivan, Data curation, Investigation; Estelle Ndukwe, Methodology; Ross Han, Sydney Cush, Muriel Lainé, Investigation; Sylvie C Mader, Conceptualization, Formal analysis, Funding acquisition, Investigation, Methodology, Project administration, Resources, Supervision, Validation, Visualization, Writing – original draft, Writing – review and editing; Geoffrey L Greene, Formal analysis, Funding acquisition, Investigation, Project administration, Resources, Supervision, Writing – review and editing; Sean W Fanning, Conceptualization, Data curation, Formal analysis, Funding acquisition, Investigation, Methodology, Project administration, Resources, Supervision, Validation, Visualization, Writing – original draft, Writing – review and editing

Author ORCIDs
Geoffrey L Greene ⬚ http://orcid.org/0000-0001-6894-8728
Sean W Fanning ⬚ http://orcid.org/0000-0002-9428-0060

Decision letter and Author response
Decision letter https://doi.org/10.7554/eLife.72512.sa1
Author response https://doi.org/10.7554/eLife.72512.sa2

# Additional files

## Supplementary files
• Transparent reporting form

## Data availability
All protein crystal structures have been deposited in the PDB under accession codes: 6PSJ, 7KBS, 7UJC, 7UJ8, 7UJM, 7UJY, 7UJF, 7UJW, 7UJO, 7UJ7, 6V8T, and 6VPF.

The following datasets were generated:

| Author(s) | Year | Dataset title | Dataset URL | Database and Identifier |
|---|---|---|---|---|
| Fanning SW, Greene GL | 2020 | Bazedoxifene in Complex with Y537S Estrogen Receptor Alpha Ligand Binding Domain | https://www.rcsb.org/structure/6PSJ | RCSB Protein Data Bank, 6PSJ |
| Fanning SW | 2020 | Estrogen Receptor Alpha Ligand Binding Domain in Complex with Raloxifene | https://www.rcsb.org/structure/7KBS | RCSB Protein Data Bank, 7KBS |
| Fanning SW, Greene GL | 2022 | Estrogen Receptor Alpha Ligand Binding Domain Y537S Mutant in Complex with Raloxifene | https://www.rcsb.org/structure/7UJC | RCSB Protein Data Bank, 7UJC |
| Fanning SW, Greene GL | 2022 | Estrogen Receptor Alpha Ligand Binding Domain Y537S in Complex with 4-Hydroxytamoxifen | https://www.rcsb.org/structure/7UJ8 | RCSB Protein Data Bank, 7UJ8 |
| Fanning SW, Greene GL | 2022 | Estrogen Receptor Alpha Ligand Binding Domain in Complex with a Methylated Lasofoxifene Derivative That Increases Receptor Resonance Time in the Nucleus of Breast Cancer Cells | https://www.rcsb.org/structure/7UJM | RCSB Protein Data Bank, 7UJM |

*Continued on next page*

*Continued*

| Author(s) | Year | Dataset title | Dataset URL | Database and Identifier |
|---|---|---|---|---|
| Fanning SW, Greene GL | 2022 | Estrogen Receptor Alpha Ligand Binding Domain Y537S Mutant in Complex with a Methylated Lasofoxifene Derivative that Enhances Estrogen Receptor Alpha Nuclear Resonance Time | https://www.rcsb.org/structure/7UJY | RCSB Protein Data Bank, 7UJY |
| Fanning SW, Greene GL | 2022 | Estrogen Receptor Alpha Ligand Binding Domain in Complex with a Methylated Lasofoxifene Derivative with Selective Estrogen Receptor Degrader Properties | https://www.rcsb.org/structure/7UJF | RCSB Protein Data Bank, 7UJF |
| Fanning SW, Greene GL | 2022 | Estrogen Receptor Alpha Ligand Binding Domain Y537S Mutant in Complex with a Methylated Lasofoxifene Derivative that Possesses Selective Estrogen Receptor Degrader Activities | https://www.rcsb.org/structure/7UJW | RCSB Protein Data Bank, 7UJW |
| Fanning SW, Greene GL | 2022 | Estrogen Receptor Alpha Ligand Binding Domain in Complex with RU39411 | https://www.rcsb.org/structure/7UJO | RCSB Protein Data Bank, 7UJO |
| Fanning SW, Greene GL | 2022 | Estrogen Receptor Alpha Ligand Binding Domain Y537S Mutant in Complex with RU39411 | https://www.rcsb.org/structure/7UJ7 | RCSB Protein Data Bank, 7UJ7 |
| Fanning SW, Greene GL | 2019 | Estrogen Receptor Alpha Ligand Binding Domain Y537S in Complex with LSZ102 | https://www.rcsb.org/structure/6V8T | RCSB Protein Data Bank, 6V8T |
| Fanning SW, Greene GL | 2020 | Estrogen Receptor Alpha Ligand Binding Domain in Complex with the Selective Estrogen Receptor Modulator Clomiphene | https://www.rcsb.org/structure/6VPF | RCSB Protein Data Bank, 6VPF |

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
