## [Editor Report]

This work elegantly provides novel insights into the molecular mechanisms of selective Estrogen Receptor modulator and downregulator activities on Estrogen Receptor function.

---

## [Decision Letter]

**Decision letter after peer review:**

Thank you for submitting your article "Stereo-specific Lasofoxifene Derivatives Reveal the Interplay between Estrogen Receptor Α Stability and Antagonistic Activity in ESR1 Mutant Breast Cancer Cells" for consideration by *eLife*. Your article has been reviewed by 2 peer reviewers, and the evaluation has been overseen by a Reviewing Editor and Anna Akhmanova as the Senior Editor. The following individual involved in review of your submission has agreed to reveal their identity: Kendall W Nettles (Reviewer #2).

This is an impressive structural and molecular mechanistic study in the modes of action of a series of Estrogen Receptor inhibitor molecules. The study is also of high relevance in the context of the resistance acquiring ER537/538 mutations. Molecular understanding of the underlying mechanisms is lacking here and the authors make a valuable contribution.

Essential revisions:

1) The authors use a halo-tagged ERα construct to quantify differences in ERα expression and cellular lifetime. Appending large protein domains to ERα potentially influences the properties of the protein. What are the data that show that this is not the case? Is the singular experiment of T47D cells with increasing doses of fulvestrant enough? This would constitute more a qualitative comparison and actually 2-3 additionally SERMs would ideally be compared in this manner as well. (Especially because this halo-tagged ER is also central to studies with the ER mutants).

2) Figure4A-C. The dose response curves look more like step-functions (Hill slope is extremely steep), than a -to be expected- curvature over at least two log decades (as is the case for example in Figure 4D-F). What is going on there?

3) The description/discussion of the crystallography data (for example movement of Helix 12) is very qualitative. Especially with such rather small movements (and not all structures at very high resolution), it would be extremely valuable (I personally would say necessary), to provide a more quantitative and statistical analysis of the data (by adding elements such as the maximum-likelihood coordinate error). Many movements in the Helix 12 position appear to be in the 0.1-0.4 Å scale and with only the refined structure shown (and not the actual electron density), there is a potential danger of overinterpretation of the data.

4) I very much appreciate that the authors selected the protein with the fewest crystal contacts in the crystal structure, to limit the effect of those contacts on the H12 as much as possible. However, it remains somewhat unclear to the reader now, for which of the structures these crystal contacts might still have influence the positioning of H12 in the final structures.

5) An interesting aspect (for future studies, potentially to be shortly discussed in the outlook of the manuscript) would be the effect/interplay of these ER modulators with the Y537 phosphorylation, which is -in the opinion of this reviewer- a highly relevant but understudied mechanism regarding responsiveness to SERM therapy. Also, because the molecular effects described for this phosphorylation are quite similar to the effects of the point mutations e.g. (a more E2 like conformation without ligand).

6) It would be helpful to know if the new compounds show SERM agonist activity, such as seen with tamoxifen in luciferase assays in the HepG2, and to a lesser extent, Ishikawa cell lines.

7) Reference 5: 50% de novo or acquired resistance at 5 years. This reference doesn't show that. See instead PMID: 30555626

8) Figure 2 is confusing. It appears that they completed time course studies (Figure 2 D-I), and then picked a time for the dose response (A-C), which are then summarized in Figure 2J. This should be reordered. The legend doesn't describe what is being done (fluorescence of halo tagged ER was assayed after addition of halo dye, dox, and ER ligand). Please reexamine the figure legends to make sure the reader can assess what was done.

9) Figure 2 presents the laso derivatives before they are explained in the text. The reader doesn't know what LA-stab and LA-Deg are at this point. These data could be removed from Figure 2A-I and Figure 2J moved to Figure 3.

10) Please renumber the chains so the A chain is the relevant chain to look at in each structure to make it easier for the viewer to look at them and superpose them. Please add a supplemental table listing all the pdb codes that were analyzed and their respective ligands.

11) Figure 7. And the associated text need significant revision.

12) Structures of the ER LBD have typically shown helix 12 in the agonist conformation, or with helix 12 flipped into the coactivator binding site to block recruitment of coactivators, which has been called the antagonist conformation. However, there is a significant body of data that antagonism involves recruitment of transcriptional corepressor proteins, which require helix 12 to be in a third position, which is unstructured. This was seen with the structure of a compound that is similar to fulvestrant, ICI 164,384, and some more recent full antagonist structures such as pdb 7rs3, 6b0f, 5T92 and 5t97. Importantly, mutations or phosphorylation of Ser305 that stabilize helix 12 in the "antagonist" conformer block corepressor recruitment and enable SERM agonist activity. The current model is that the degree of helix 12 destabilization drives efficacy, and that the so called "antagonist conformation" is actually the SERM agonist conformer by blocking corepressors, and allowing the amino terminal coactivator binding domain to have some constitutive activity see PMID: 34452998, 17283072, 1250199, 12482846. See also Dr. Fanning's previous *eLife* paper showing that the efficacy of bazedoxifene is through pushing on the helix 11-12 loop to destabilize helix 12.

Given this background, the conclusion of the structural studies was suggested to support the opposite conclusion, that stabilizing helix 12 leads to greater efficacy. After examining the structures in detail there does not appear to be a clear SAR between luciferase activity and helix 12 position, and it was not clear that the luciferase data correlated with anti-proliferative maximal efficacy.

13) The authors should reexamine the structures to identify how Ser537 and Tyr537 elicit different effects on the helix 11-12 loop interaction with the ligand to drive different pharmacology.

14) in the Baz 537S structure, the loop position is due to crystal packing, which could pull helix 12 and this should be noted or these structures not discussed.

15) The positions of amino acids 537, 373, and 351 should be reexamined in the structures. There are obvious errors in 7bk5 that if real space refined show the H-bond between Tyr537 and His373

16) The discussion and introduction have some redundancy. It might be better to move the extensive analysis on why degradation is not important for efficacy to the discussion and just briefly mention in the intro that there is increasing evidence and give the references. Sentences on line 439 and 440 are not needed in the discussion.

---

## [Author Response]

Essential revisions:The authors use a halo-tagged ERα construct to quantify differences in ERα expression and cellular lifetime. Appending large protein domains to ERα potentially influences the properties of the protein. What are the data that show that this is not the case? Is the singular experiment of T47D cells with increasing doses of fulvestrant enough? This would constitute more a qualitative comparison and actually 2-3 additionally SERMs would ideally be compared in this manner as well. (Especially because this halo-tagged ER is also central to studies with the ER mutants).

We have performed dose-response curves using an in-cell western blot in T47D breast cancer cell lines treated with 4-hydroxytamoxifen (4OHT), fulvestrant (ICI), lasofoxifene (laso), laso-degrader (LA-Deg), laso-stablizer (LA-Stab). This assay measures endogenous estrogen receptor within the T47D cells and is normalized to cell number. Overall, these data agree with the halo-tag assay. Given the validation with the earlier fulvestrant study and that our data agree with previously published results, these data show that the halo-tag assay accurately recapitulates endogenous ERα responses to hormone, SERM, and SERD treatment in breast cancer cells^1,2^.

The description/discussion of the crystallography data (for example movement of Helix 12) is very qualitative. Especially with such rather small movements (and not all structures at very high resolution), it would be extremely valuable (I personally would say necessary), to provide a more quantitative and statistical analysis of the data (by adding elements such as the maximum-likelihood coordinate error). Many movements in the Helix 12 position appear to be in the 0.1-0.4 Å scale and with only the refined structure shown (and not the actual electron density), there is a potential danger of overinterpretation of the data.I very much appreciate that the authors selected the protein with the fewest crystal contacts in the crystal structure, to limit the effect of those contacts on the H12 as much as possible. However, it remains somewhat unclear to the reader now, for which of the structures these crystal contacts might still have influence the positioning of H12 in the final structures.Please renumber the chains so the A chain is the relevant chain to look at in each structure to make it easier for the viewer to look at them and superpose them. Please add a supplemental table listing all the pdb codes that were analyzed and their respective ligands.Figure 7. And the associated text need significant revision.The authors should reexamine the structures to identify how Ser537 and Tyr537 elicit different effects on the helix 11-12 loop interaction with the ligand to drive different pharmacology.

We have reanalyzed the crystal structures and have made significant changes to the associated text and figures. Because they were relatively minor, we have removed the discussion of ligand-specific changes and have instead focused the section on the clear structural features that are observed in the most potent molecules. In terms of qualitative discussion, we originally calculated buried surface area for each helix 12 in the Y537S structures and observed a trend where the molecules that favored the most burial in the AF2 cleft also showed the greatest anti-transcriptional efficacies. In the revised manuscript, we have now also included B-factor analysis and show that the H12 in the Y537S structures that has the lowest B-factor (normalized to the average B-factor across the structure) are those that show the greatest anti-transcriptional potencies.

We made a new Figure 7. Instead of showing every structure, we now highlight the structural features of the most versus least effective molecules in the MCF7 WT/Y537S *ESR1* cells. Specifically, we observe that the most effective molecules show little difference versus their WT counterpart, form a hydrogen bond between S537 and E380, and the 2mFo-DFc difference maps of H12 are rigid and well packed in the AF-2 cleft. The least effective molecules lack this hydrogen bond and are poorly packed.

Further descriptions of the crystal contact are included in the results. A supplemental figure has been included that highlights the crystal contact formed between D538 and a symmetry mate arginine affects H12 and disqualified those monomers from the analysis. We have also renamed the chains so that chain A is the chain of interest in these structures.

An interesting aspect (for future studies, potentially to be shortly discussed in the outlook of the manuscript) would be the effect/interplay of these ER modulators with the Y537 phosphorylation, which is -in the opinion of this reviewer- a highly relevant but understudied mechanism regarding responsiveness to SERM therapy. Also, because the molecular effects described for this phosphorylation are quite similar to the effects of the point mutations e.g. (a more E2 like conformation without ligand).

We agree and have added in some text in the discussion about the importance of Y537 phosphorylation as well as potential changes to which coregulators associate with mutant versus WT ERα in the presence of SERMs and SERDs.

It would be helpful to know if the new compounds show SERM agonist activity, such as seen with tamoxifen in luciferase assays in the HepG2, and to a lesser extent, Ishikawa cell lines.

We attempted to study SERM agonist activity using HepG2 cells either endogenously or by transfecting ERα and were unable to measure meaningful responses using a reporter gene system. We were able to obtain these data in Ishikawa cells purchased from Σ Aldrich as well as ECC1 cells that were donated by Olema Oncology. We also purchased Ishikawa cells from ATCC but had difficulty culturing them. We used an established alkaline phosphatase activity assay to measure ERα agonism in these cells.^3^ These results show that the SERM-like LA-Stab does show SERM-agonistic activities in these assays. Laso showed a slight induction, while LA-Deg (the SERD) did not induce activity. These experiments were performed alongside 4OHT, ICI, and Z-endoxifen. Interestingly, E2 showed different degrees of stimulation between the two cells lines. However, the trend in SERM-agonism was the same in both cell lines.

Reference 5: 50% de novo or acquired resistance at 5 years. This reference doesn't show that. See instead PMID: 30555626

Citation has been changed.

Figure 2 is confusing. It appears that they completed time course studies (Figure 2 D-I), and then picked a time for the dose response (A-C), which are then summarized in Figure 2J. This should be reordered. The legend doesn't describe what is being done (fluorescence of halo tagged ER was assayed after addition of halo dye, dox, and ER ligand). Please reexamine the figure legends to make sure the reader can assess what was done.Figure 2 presents the laso derivatives before they are explained in the text. The reader doesn't know what LA-stab and LA-Deg are at this point. These data could be removed from Figure 2A-I and Figure 2J moved to Figure 3.

The text and figure 2 have been reordered to discuss the time-course studies first then the dose-response and have revised the legend to be more descriptive. All LA-Stab and LA-Deg time-course studies into Figure 3. The summary figure panel has also been moved to Figure 3.

Structures of the ER LBD have typically shown helix 12 in the agonist conformation, or with helix 12 flipped into the coactivator binding site to block recruitment of coactivators, which has been called the antagonist conformation. However, there is a significant body of data that antagonism involves recruitment of transcriptional corepressor proteins, which require helix 12 to be in a third position, which is unstructured. This was seen with the structure of a compound that is similar to fulvestrant, ICI 164,384, and some more recent full antagonist structures such as pdb 7rs3, 6b0f, 5T92 and 5t97. Importantly, mutations or phosphorylation of Ser305 that stabilize helix 12 in the "antagonist" conformer block corepressor recruitment and enable SERM agonist activity. The current model is that the degree of helix 12 destabilization drives efficacy, and that the so called "antagonist conformation" is actually the SERM agonist conformer by blocking corepressors, and allowing the amino terminal coactivator binding domain to have some constitutive activity see PMID: 34452998, 17283072, 1250199, 12482846. See also Dr. Fanning's previous eLife paper showing that the efficacy of bazedoxifene is through pushing on the helix 11-12 loop to destabilize helix 12.Given this background, the conclusion of the structural studies was suggested to support the opposite conclusion, that stabilizing helix 12 leads to greater efficacy. After examining the structures in detail there does not appear to be a clear SAR between luciferase activity and helix 12 position, and it was not clear that the luciferase data correlated with anti-proliferative maximal efficacy.

SERM/SERD-induced AF-2 antagonism is complex and stems from multiple contributing context/tissue-dependent influences on estrogen receptor α. Helix 12 in the AF-2 cleft occludes co-activator (i.e. SRC3) binding, while recruitment of corepressor enzymes like NCOR or SMRT to ERα also contribute epigenetic silencing activities^4^. 4-hydroxtamoxifen (4OHT) can induce ERα-NCOR binding but fulvestrant and raloxifene do so more efficiently^4^. These studies suggest H12 instability enhances co-repressor recruitment, which improves anti-transcriptional activities. It is also established that H12 instability correlates with ERα-degrading activities. Therefore, SERD-induced ERα-degradation should predict co-repressor recruitment and enhanced anti-transcriptional therapeutic activities. However, studies from the McDonnell lab suggests that the fulvestrant therapeutic activities stem from ERα transcriptional antagonism to a greater degree than degradation^5^. Further, lasofoxifene, a relatively poor ERα degrader, demonstrates improved activities over fulvestrant in MCF7 cells with WT or Y537S ERα^6^. Together, these studies suggest that, in breast cancer cells, ERα transcriptional antagonism may not correlate strongly with the degree of receptor degradation. In our study, we synthesized the lasofoxifene derivatives in order to test this hypothesis. These molecules, along with a panel of SERMs, SERM/SERDs, and SERDs suggest that degradation does not predict transcriptional antagonistic potency in breast cancer cell lines harboring WT or WT/mutant *ESR1* under our assay conditions. As discussed, it may play a role in partial agonist activity of antiestrogens in a tissue- or possibly gene-specific specific manner.

Another major area of this study is revealing the structural basis for differential SERM or SERD activities for Y537S ERα. Structural studies were performed in parallel with the biological studies and we sought to solve as many structures as possible to identify structural features that correlate with efficacy. Our previous bazedoxifene (BZA) study did suggest that mobilizing H12 improved activity for Y537S ERα compared to 4OHT. However, we were unable to solve a structure of the molecule in complex with the Y537S mutant. Rather, we used solution-phase protein dynamics studies using HDX-MS of BZA in complex with WT or Y537S ERα LBD. We also performed detailed atomistic molecular dynamics simulations of the BZA bound to Y537S ERα LBD. Both studies from that paper agree with our current structural analysis. The HDX-MS data showed that H12 is less dynamic in the Y537S-BZA structure versus WT. The MD-sim predicted the formation of the S537-E380 hydrogen bond.

In terms of the structures of ICI 164,380 and other pure antagonist SERDs. In the PDB, the ICI 164,380 structure is with rat ERβ and lacks an ordered H12. We recently solved a structure of the same ligand and derivatives thereof in complex with ERα LBD carrying the mutation L536S in collaboration with our co-author Dr. Sylvie Mader for a forthcoming paper. Here, ERα H12 is observed in the antagonistic conformation in the AF-2 cleft but it is less helical than what would be observed in a SERM structure (see Author response image 1). Modeling of fulvestrant binding with S536 or L536 LBDs also maintained H12 in antagonist conformations. As for the other ligands, we have evaluated the models and the maps and at least some difference density is observed in the AF-2 cleft in the monomers that lack a modeled H12. While it is not significant in all of the structures, it is readily apparent in the 5T92 and 5T97 (see Author response image 1). In our experience, additional rounds of refinement likely would have led to a better-observed H12 and would have enabled the modeling of at least the main-chain. These structures suggest that H12 is not completely in a new disordered conformation but may still pack into the AF-2 cleft.

**Author response image 1. sa2fig1:** 

We lack a complete understanding of how SERMs or SERDs induce ERα corepressor binding and the role these protein play in antagonistic activities across multiple hormone-responsive tissues. The NCOR-ERα binding study from the Kushner lab^4^ suggests that H12 must be absent from the AF2 cleft for full binding. They also show that the hinge region, in between the DNA-binding domain and LBD, is important for NCOR binding. Likewise, a decreased co-repressor recruitment is observed with phospho S305, which lies near the ERα hinge-region far from the AF-2 cleft^7^. Together, these studies suggest that co-repressors likely bind along multiple surfaces on ERα beyond just the AF-2 cleft. Recent, cryo-EM structures of ERα and AR from the O’Malley lab also show significant contacts between AF-2 co-activators and the respective LBDs^8,9^. Additionally, a recent study from the Nettles and Katzenellenbogen laboratories shows that ERα antagonists that affect helix 8, in addition to helix 12, alters the repertoire of associated coregulators and enhances anti-cancer activities^10^. Together, these studies show that coregulator binding occurs across multiple surfaces of ERα and more studies are needed to improve our understanding of these relationships.

14) in the Baz 537S structure, the loop position is due to crystal packing, which could pull helix 12 and this should be noted or these structures not discussed.

This is true for Chain B, which shows a potential crystal contact at L536 as well a hydrogen bond formed between D538 and R436 of a symmetry mate. We chose to analyze Chain A in this structure because there are no crystal contacts observed until the c-terminus of the protein. There is a symmetry mate that approaches the H11-12 loop around residues 530 to 532. However, there is no clear contact formed here and the closest it reaches is 3.8 Å, outside hydrogen bonding distance. Crystal contacts are frequent confounding factors in crystal structures that, in our opinion, are not discussed as often enough. We have included additional descriptions of the crystal contacts in the Results section.

15) The positions of amino acids 537, 373, and 351 should be reexamined in the structures. There are obvious errors in 7bk5 that if real space refined show the H-bond between Tyr537 and His373

We assume that the Reviewer is referring to 7KBS. Each of these structures have been extensively refined but we have reanalyzed the positioning of 537, 373, and 351. We have redeposited the coordinates as needed. As is the case with many ERα LBD structures, Y537 is not fully ordered in each chain, but is structured enough for us to have some confidence in its positioning. For 7KBS, we solved this structure because we wanted to improve upon the 2.6 Å of PDB: 1ERR. Indeed, we were able to solve the structure of RAL in complex with ERα LBD to 1.83 Å and in the same C2 spacegroup as 1ERR. In the 1ERR structure, Y537 is ordered in Chain A but not in Chain B and is poised for a hydrogen bond with H373. However, the density around H373 is poorly resolved in both monomers of 1ERR. In 7KBS, Y537 and H373 are better resolved in both chains. While Y537 is not fully resolved in Chain A, the curvature of the density suggests that we have modeled it in the most representative side-chain conformation. Neither monomer suggests the presence of an ideal hydrogen bond, given the roughly 90^o^ angle between the phenol and imidazole. In sum, there is a potential hydrogen bond between Y537 and H373 in 1ERR but the sidechains are poorly resolved, while the sidechains better resolved in 7KBS and do not show the formation of a hydrogen bond.

References1. Zhao, Y. *et al.* Structurally Novel Antiestrogens Elicit Differential Responses from Constitutively Active Mutant Estrogen Receptors in Breast Cancer Cells and Tumors. *Cancer research* 77, 5602-5613, doi:10.1158/0008-5472.CAN-17-1265 (2017).

2. Fanning, S. W. *et al.* The SERM/SERD bazedoxifene disrupts ESR1 helix 12 to overcome acquired hormone resistance in breast cancer cells. *eLife* 7, e37161, doi:10.7554/*eLife*.37161 (2018).

3. Fanning, S. W. *et al.* Specific stereochemistry of OP-1074 disrupts estrogen receptor α helix 12 and confers pure antiestrogenic activity. *Nature Communications* 9, 2368, doi:10.1038/s41467-018-04413-3 (2018).

4. Webb, P., Nguyen, P. and Kushner, P. J. Differential SERM effects on corepressor binding dictate ERalpha activity in vivo. *J Biol Chem* 278, 6912-6920, doi:10.1074/jbc.M208501200 (2003).

5. Wardell, S. E. *et al.* Pharmacokinetic and pharmacodynamic analysis of fulvestrant in preclinical models of breast cancer to assess the importance of its estrogen receptor-α degrader activity in antitumor efficacy. *Breast Cancer Res Treat* 179, 67-77, doi:10.1007/s10549-019-05454-y (2020).

6. Lainé, M. *et al.* Lasofoxifene as a potential treatment for therapy-resistant ER-positive metastatic breast cancer. *Breast Cancer Research* 23, 54, doi:10.1186/s13058-021-01431-w (2021).

7. Houtman, R. *et al.* Serine-305 phosphorylation modulates estrogen receptor α binding to a coregulator peptide array, with potential application in predicting responses to tamoxifen. *Mol Cancer Ther* 11, 805-816, doi:10.1158/1535-7163.Mct-11-0855 (2012).

8. Yu, X. *et al.* Structural Insights of Transcriptionally Active, Full-Length Androgen Receptor Coactivator Complexes. *Molecular Cell* 79, 812-823.e814, doi:https://doi.org/10.1016/j.molcel.2020.06.031 (2020).

9. Yi, P. *et al.* Structure of a biologically active estrogen receptor-coactivator complex on DNA. *Molecular cell* 57, 1047-1058, doi:10.1016/j.molcel.2015.01.025 (2015).

10. Min, J. *et al.* Dual-mechanism estrogen receptor inhibitors. *Proceedings of the National Academy of Sciences* 118, e2101657118, doi:10.1073/pnas.2101657118 (2021).